# The rapidly evolving X-linked *MIR-506* family fine-tunes spermatogenesis to enhance sperm competition

Zhuqing Wang[1,2], Yue Wang[1,2], Tong Zhou[1], Sheng Chen[1,2], Dayton Morris[2], Rubens Daniel Miserani Magalhães[2], Musheng Li[1], Shawn Wang[1], Hetan Wang[1,2], Yeming Xie[1], Hayden McSwiggin[1,2], Daniel Oliver[1], Shuiqiao Yuan[1], Huili Zheng[1,2], Jaaved Mohammed[3], Eric C Lai[3], John R McCarrey[4], Wei Yan[1,2,5]*

[1]Department of Physiology and Cell Biology, University of Nevada, Reno School of Medicine, Reno, United States; [2]The Lundquist Institute for Biomedical Innovation at Harbor-UCLA Medical Center, Torrance, United States; [3]Developmental Biology Program, Sloan Kettering Institute, New York, United States; [4]Department of Neuroscience, Developmental and Regenerative Biology, University of Texas at San Antonio, San Antonio, United States; [5]Department of Medicine, David Geffen School of Medicine, University of California, Los Angeles, Los Angeles, United States

*For correspondence:
wei.yan@lundquist.org

**Abstract** Despite rapid evolution across eutherian mammals, the X-linked *MIR-506* family miRNAs are located in a region flanked by two highly conserved protein-coding genes (*SLITRK2* and *FMR1*) on the X chromosome. Intriguingly, these miRNAs are predominantly expressed in the testis, suggesting a potential role in spermatogenesis and male fertility. Here, we report that the X-linked *MIR-506* family miRNAs were derived from the MER91C DNA transposons. Selective inactivation of individual miRNAs or clusters caused no discernible defects, but simultaneous ablation of five clusters containing 19 members of the *MIR-506* family led to reduced male fertility in mice. Despite normal sperm counts, motility, and morphology, the KO sperm were less competitive than wild-type sperm when subjected to a polyandrous mating scheme. Transcriptomic and bioinformatic analyses revealed that these X-linked *MIR-506* family miRNAs, in addition to targeting a set of conserved genes, have more targets that are critical for spermatogenesis and embryonic development during evolution. Our data suggest that the *MIR-506* family miRNAs function to enhance sperm competitiveness and reproductive fitness of the male by finetuning gene expression during spermatogenesis.

## eLife assessment

This study provides **important** findings on the evolution and function of the X-linked *MIR-506* family. The evidence supporting the conclusions is **convincing**, including the generation and characterization of an impressive number of the miRNA deletion mutants. This work will be of interest to RNA biologists, evolution biologists, and reproductive biologists.

## Introduction

Spermatogenesis is highly conserved among all vertebrates. Although it generally consists of three phases (mitotic, meiotic, and haploid), many characteristics appear to be species-specific, for example, the duration of each of the three phases, the seminiferous epithelial organization, and the shape and length of spermatozoa, likely reflecting the adaptive changes during evolution (***Oakberg, 1957***;

*Muciaccia et al., 2013*; *Gu et al., 2019*). Several cellular events are unique to the male germ cells, for example, postnatal formation of the adult male germline stem cells (i.e., spermatogonia stem cells), pubertal onset of meiosis, and haploid male germ cell differentiation (also called spermiogenesis) (*Hermo et al., 2010*). Unique cellular processes are often accompanied by a more complex yet unique transcriptome, which may explain why the testis expresses more genes than any other organs in the body, with the possible exception of the brain (*Khaitovich et al., 2006*). Regulation of gene expression during spermatogenesis occurs at both transcriptional and post-transcriptional levels (*Idler and Yan, 2012*). As a post-transcriptional regulator, miRNAs are abundantly expressed in the testis and are required for spermatogenesis (*Ro et al., 2007*; *Wu et al., 2012*; *Papaioannou et al., 2009*; *Zhang et al., 2017*; *Guo et al., 2022*). miRNAs typically function at post-transcriptional levels by binding the complementary sequences in the untranslated regions (UTRs) of mRNAs – particularly in the 3′UTRs through the 'seed sequence' (2nd–7th nucleotides)(*Bartel, 2009*). Numerous miRNAs are subject to rapid evolution, probably in response to the accelerated rate of divergence of UTRs compared to the exonic sequences (*Mayr, 2016*). Divergence of genomic sequences can be mediated by transposable elements (TEs), which are known as building blocks of the genome and mostly map to UTRs and intronic regions of protein-encoding genes (*Thompson et al., 2016*). Each miRNA can bind numerous target mRNAs, and one mRNA can be targeted by multiple different miRNAs. This 'one-to-multiple' relationship between miRNAs and mRNAs amplifies their potential to coordinate gene expression in the cell (*Bartel, 2009*). Moreover, miRNA genes often exist in clusters, which are transcribed as a unit followed by nuclear and cytoplasmic cleavage events to generate individual miRNAs (*Bartel, 2009*).

Multiple clusters of miRNA genes containing the same seed sequences are categorized as a miRNA family, and miRNAs within the same family likely evolved from a common ancestor sequence (*Wang et al., 2020b*). Of great interest, many of the testis-enriched miRNA clusters map to the X chromosome (*Song et al., 2009*). Sex-linked genes are generally subject to the male germline-specific phenomenon called meiotic sex chromosome inactivation (MSCI), which silences transcription during most, if not all, of meiosis (*Song et al., 2009*). Indeed, prior to 2009, there were no confirmed reports of any sex-linked genes escaping the repressive effects of MSCI. Surprisingly, however, we found that many X-linked miRNA genes do escape MSCI, suggesting that they may contribute to particularly important functions during spermatogenesis (*Song et al., 2009*). This notion has since been tested by the generation of knockouts (KOs) of individual miRNA genes or individual clusters of miRNA genes normally expressed during spermatogenesis. However, these KOs resulted in minimal, if any, phenotypic effects and did not appear to impede normal spermatogenesis or male fertility (*Wang et al., 2020b*). This left the field facing multiple unanswered questions, including (1) why do so many X-linked miRNAs express uniquely or preferentially during spermatogenesis and escape MSCI, (2) what is their origin, and (3) how and why did they evolve rapidly?

To address these questions and better understand the functional role played by these X-linked miRNAs, we investigated the evolutionary history of this unique miRNA family and also generated KOs of individual, paired, triple, quadruple, or quintuple sets of miRNA clusters within this family and tested the effects on male fertility, initially by standard monandrous mating assays. Consistent with previous efforts to inactivate miRNA genes in *Caenorhabditis elegans* and mice (*Wang et al., 2020b*; *Bao et al., 2012*; *Wang et al., 2022*; *Miska et al., 2007*), KOs of either individual members or individual clusters of the *MIR-506* family induced no discernible phenotypes and did not impact male fertility. This may reflect the level of functional redundancy inherent within the members and clusters of this miRNA family. It was only when four or more clusters of the *MIR-506* family were ablated that relevant phenotype became detectable, which was manifested in the form of reduced litter size despite normal sperm counts, motility, and morphology. Interestingly, the most common male fertility testing for lab rodents is based on a monandrous mating scheme, that is, one fertility-proven female mated with one male. However, there are additional aspects of male fertility in many mammalian species, particularly those that are normally litter-bearing. In the wild, litter-bearing females often mate with multiple different males such that a single litter may include pups sired by more than one male (*Dean et al., 2006*; *Firman and Simmons, 2008*). This polyandrous mating introduces the potential for additional aspects of male reproductive fitness to accrue, one of which involves sperm competition (*Dean et al., 2006*; *Firman and Simmons, 2008*). Sperm competition can occur when sperm from more than one male are present in the female reproductive tract simultaneously, such that they then compete to fertilize each oocyte (*Parker, 1970*). Sperm competition has been now recognized as a significant

evolutionary force directly impacting male reproductive success (*Parker, 1970*). Using experiments that mimic polyandrous mating, we found that the quinKO male mice indeed displayed compromised sperm competition. Hence, the X-linked *MIR-506* family miRNAs appear to function to finetune spermatogenesis to enhance sperm competition and, consequently, male reproductive fitness.

## Results

### X-linked *MIR-506* family miRNAs flanked by two highly conserved protein-coding genes *SLITRK2* and *FMR1* rapidly evolved across species

X-linked genes are generally more divergent between species than autosomal ones, a phenomenon known as the 'faster-X effect' (*Meisel and Connallon, 2013*). However, despite a high degree of conservation of two protein-coding genes, *SLITRK2* and *FMR1*, on the X chromosome across species, the miRNA genes located between these two loci are divergent among clades across the eutherian mammals (*Wang et al., 2020b*; *Zhang et al., 2019*). Through tracing the evolution of this genomic region, we found that *SLITRK2* and *FMR1* mapped to chromosome 4 (syntenic with mammalian X chromosome) in zebrafish and birds, but to the X chromosome in most mammals, with divergence and multiplication of numerous miRNA genes that belong to the *MIR-506* family in between (*Figure 1A* and *Supplementary file 1*). By mapping these miRNAs of various species using the UCSC genome browser (*Casper et al., 2018*), we found that all members of the *MIR-506* family are located in a region flanked by *SLITRK2* and *FMR1* (*Figure 1A*). Consistent with previous reports (*Wang et al., 2020b*; *Zhang et al., 2019*), *SLITRK2* and *FMR1* are usually on the positive strand, whereas the *MIR-506* family miRNAs are in the reverse orientation (*Figure 1A*). Based on the location of these miRNAs, we named the miRNAs proximal to *SLITRK2* (*MIR892C~MIR891A* in humans) and *FMR1* (*MIR513C~MIR514A3* in humans) *SmiRs* (*SLITRK2*-proximal miRNAs) and *FmiRs* (*FMR1*-proximal miRNAs), respectively.

To evaluate the sequence conservation of these miRNAs across species, we adopted the Multiz Alignment and Conservation pipeline, which utilizes PhastCons and PhyloP algorithms (*Casper et al., 2018*), to search miRNA datasets from 100 different species using the human genome as a reference (*Figure 1B*, *Figure 1—figure supplement 1*). The mean values of PhyloP and PhastCons of the *FMR1* and *SLITRK2* coding sequences (CDS) were ~4.9 and ~0.97, respectively (*Figure 1C and D*), indicating that these regions are highly conserved. In contrast, the mean values of PhyloP and PhastCons of *SmiRs* were ~1.0 and ~0.002, respectively, and those of *FmiRs* are ~0.03 and ~0.11, respectively (*Figure 1C and D*). The PhyloP and PhastCons values of the CDS, *FmiRs*, and *SmiRs* are significantly different from each other (adjusted p-value <0.05, Kruskal–Wallis test) (*Figure 1C and D*), indicating that *FmiRs* and *SmiRs* are highly divergent, and *SmiRs* are more divergent than *FmiRs*.

We then assessed the genomic sequence similarity among various species using D-GENIES-based dot plot analyses (*Cabanettes and Klopp, 2018*). Although the *SLITRK2-FMR1* genomic regions were highly variable among different species, the sequences within some clades shared a high degree of similarities, for example, primates (rhesus monkeys, chimpanzees, and humans), *cetartiodactyla* (sheep and cows), *rodentia* (e.g., mice and rats), and *carnivora* (e.g., dogs and cats) (*Figure 1—figure supplement 2A*). Although the *MIR-506* family miRNAs were highly divergent, some orthologs displayed a higher degree of sequence conservation (*Figure 1—figure supplement 2B and C* ), for example, the *SmiRs* within primates were similar (*Figure 1—figure supplement 2B*); *MIR891A* and *MIR891B* in rhesus monkeys were similar to *MIR891B* and *MIR891A* in humans and chimpanzees, respectively, and *MIR892B* in chimpanzees was homologous to *MIR892C* in humans and rhesus monkeys in terms of sequence and location (*Figure 1—figure supplement 2B*). The *FmiRs*, including *Mir201* (assigned as *Mir-506-P1* [paralogue 1] in MirGeneDB; *Fromm et al., 2020*), *Mir547* (*Mir-506-P2*), and *Mir509* (*Mir-506-P7*) in mice and rats are orthologs of *MIR506*, *MIR507*, and *MIR509* in humans, respectively (*Figure 1—figure supplement 2C* and *Supplementary file 1*). Of interest, although the *MIR-506* family miRNAs are highly divergent, the seed sequences of some miRNAs, such as *Mir-506-P6* and *Mir-506-P7*, remain conserved (*Figure 1—figure supplement 2C*), and these miRNAs represent the dominant mature miRNAs (*Kozomara et al., 2019*). It is noteworthy that the majority of the substitutions among the *MIR-506* family are U→C and A→G (*Figure 1—figure supplement 2B and C*). Furthermore, we analyzed the conservation of the *MIR-506* family in modern humans using data from the 1000 Genomes Project (1kGP) (*Figure 1E and F*; *Byrska-Bishop et al., 2022*). We compared

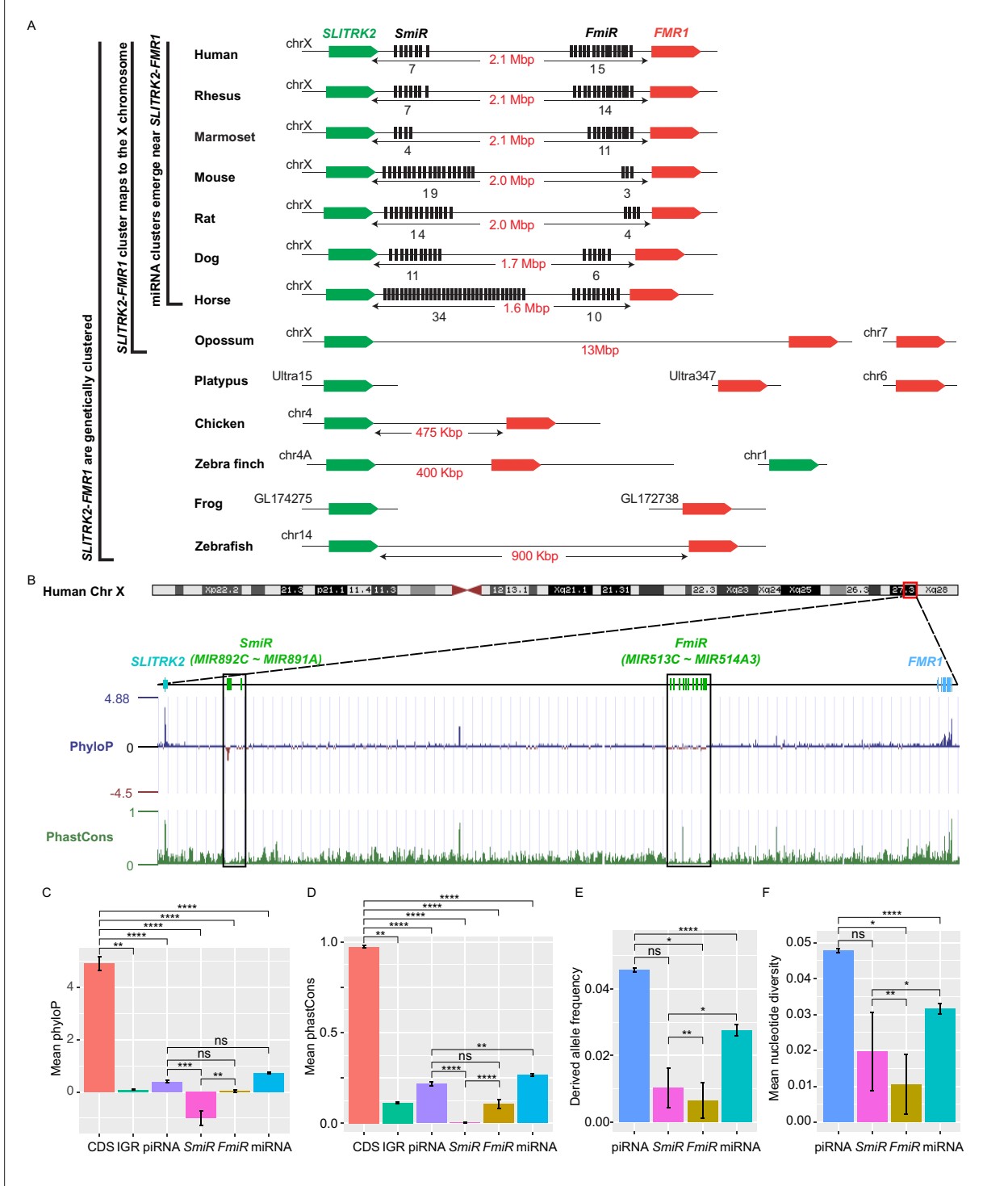

**Figure 1.** Genomic location, sequence alignment, and evolution conservation of the X-linked *MIR-506* family. (**A**) Genomic location of the X-linked *MIR-506* family miRNAs (black bars) and the two flanking coding genes, *SLITRK2* (green blocks) and *FMR1* (red blocks). The number of miRNAs within each cluster is indicated underneath the miRNA clusters. (**B**) Evolution conservation of X-linked *MIR-506* family based on Multiz Alignment and Conservation using the human genome as a reference. Positive PhyloP scores indicate conservation and vice versa. PhastCons has a score between 0–1, and the higher the score, the more conserved the DNA region is. (**C, D**) Comparison of mean PhyloP (**C**) and PhastCons (**D**) scores among CDS of *SLITRK2* and *FMR1*, intergenic region (IGR), pachytene piRNAs, *SmiRs*, *FmiRs*, and all miRNAs. **, ***, and **** indicate adjusted p-value <0.01, 0.001, and 0.0001, respectively. ns, not significant. Kruskal–Wallis test was used for statistical analyses. (**E, F**) Comparison of derived allele (**E**) and mean nucleotide

*Figure 1 continued on next page*

Figure 1 continued

(**F**) frequencies among pachytene piRNAs, *SmiRs*, *FmiRs*, and all miRNAs. *, ** and **** indicate adjusted p-value <0.05, 0.01, and 0.0001, respectively. ns, not significant. Kruskal–Wallis test was used for statistical analyses.

The online version of this article includes the following figure supplement(s) for figure 1:

**Figure supplement 1.** Multiz Alignment and Conservation analyses of X-linked *MIR-506* family across 100 species using the human genomes as references.

**Figure supplement 2.** Genomic and sequence similarity among members of the X-linked *MIR-506* family.

*SmiRs*, *FmiRs*, and all miRNAs with pachytene piRNAs, which are known to be highly divergent in modern humans but barely exert any biological functions (*Özata et al., 2020*). Of interest, the derived allele frequency (DAF) and mean nucleotide diversity (MND) of *FmiRs* and all miRNAs were significantly smaller than that of the pachytene piRNAs, whereas *SmiRs* were significantly smaller than all miRNAs (*Figure 1E and F*) (adjusted p-value<0.05, Kruskal–Wallis test), suggesting that the *MIR-506* family miRNAs are more conserved than pachytene piRNAs in modern humans. Taken together, these data indicate that the X-linked *MIR-506* family, although rapidly evolving as a whole, contains both divergent and conserved miRNAs, suggesting both conserved and novel functions across species.

## X-linked *MIR-506* family miRNAs are derived from MER91C DNA transposons

To visualize the family history of these miRNAs, we built a phylogram for the *MIR-506* family (*Figure 2—figure supplement 1*). The phylogram suggests that these miRNAs shared a common ancestor and that the *FmiRs* emerged earlier than the *SmiRs*, which is also supported by the fact that some *FmiRs* exist in green sea turtles (*Figure 1—figure supplement 1*, *Figure 2—figure supplement 1*). These data suggest that the *MIR-506* family miRNAs arose much earlier than previously thought (*Zhang et al., 2019*). The two subfamilies, *FmiRs* and *SmiRs*, despite their common ancestors, may have evolved at different paces and thus, might be functionally divergent.

Studies have shown that TEs drive evolution through transpositions (*Fedoroff, 2012*). CRISPR-Cas9/ Cas12a genome editing can induce irreversible small indels at the cutting sites (also called 'scars') (*Wang et al., 2020a*), which have been used for lineage tracing (*McKenna et al., 2016*). Inspired by this strategy, we attempted to trace the evolution of the *MIR-506* family miRNAs by searching the transposon database for the transpositional 'scars' (partial TE sequences) after transposition. To search the potential TE sources of the *MIR-506* family miRNAs, we downloaded all transposons in the human, horse, dog, and guinea pig genomes and aligned them to their corresponding *MIR-506* family miRNAs using BLAST (Basic Local Alignment Search Tool) (*Altschul et al., 1990*). The nonautonomous MER91C DNA transposon (~100–150 million years) (*Giordano et al., 2007*; *Pace and Feschotte, 2007*) was the only transposon that aligned to *FmiRs* of the *MIR-506* family (>94% identical matches) in all four species (*Supplementary file 2*).

Given that the *FmiRs* (e.g., human *MIR506~509*) emerged much earlier than the *SmiRs* (*Figure 1— figure supplement 1*, *Figure 2—figure supplement 1*) and that human *MIR513* (belonging to *FmiRs*) and *SmiRs* (including human *MIR892A* and *MIR892B*) share a common ancestor (*Figure 2—figure supplement 1*), we reasoned that the X-linked *MIR-506* family might be derived from the MER91C DNA transposon. To test this hypothesis, we first aligned the X-linked *MIR-506* family miRNAs from several species to a human MER91C DNA transposon. Indeed, numerous *FmiRs* of almost all species analyzed aligned to the MER91C DNA transposon despite few mismatches (*Figure 2A*). The phylogenetic tree further confirmed that the MER91C is the sister group of the *MIR-506* family miRNAs (*Figure 2B*, *Figure 2—figure supplement 2*), suggesting that the MER91C DNA transposon is the likely source of the older *MIR-506* family miRNAs. Further supporting this notion, the MER91C DNA transposons could form hairpin structures, which is a prerequisite for miRNA biogenesis (*Figure 2C*, *Figure 2—figure supplement 3A*). Moreover, analyses of the testis small RNA datasets from humans, marmosets, dogs, and horses revealed the peaks corresponding to these miRNAs (*Figure 2D*, *Figure 2—figure supplement 3A*). Finally, by overexpressing several MER91C DNA regions randomly selected from humans, dogs, and horses in HEK293T cells, we found that these DNA regions were indeed capable of producing miRNAs (*Figure 2E and F*, *Figure 2—source data 1 and 2* and *Figure 2—figure supplement 3B, C*, and *Figure 2—figure supplement 3—source data 1 and 2*).

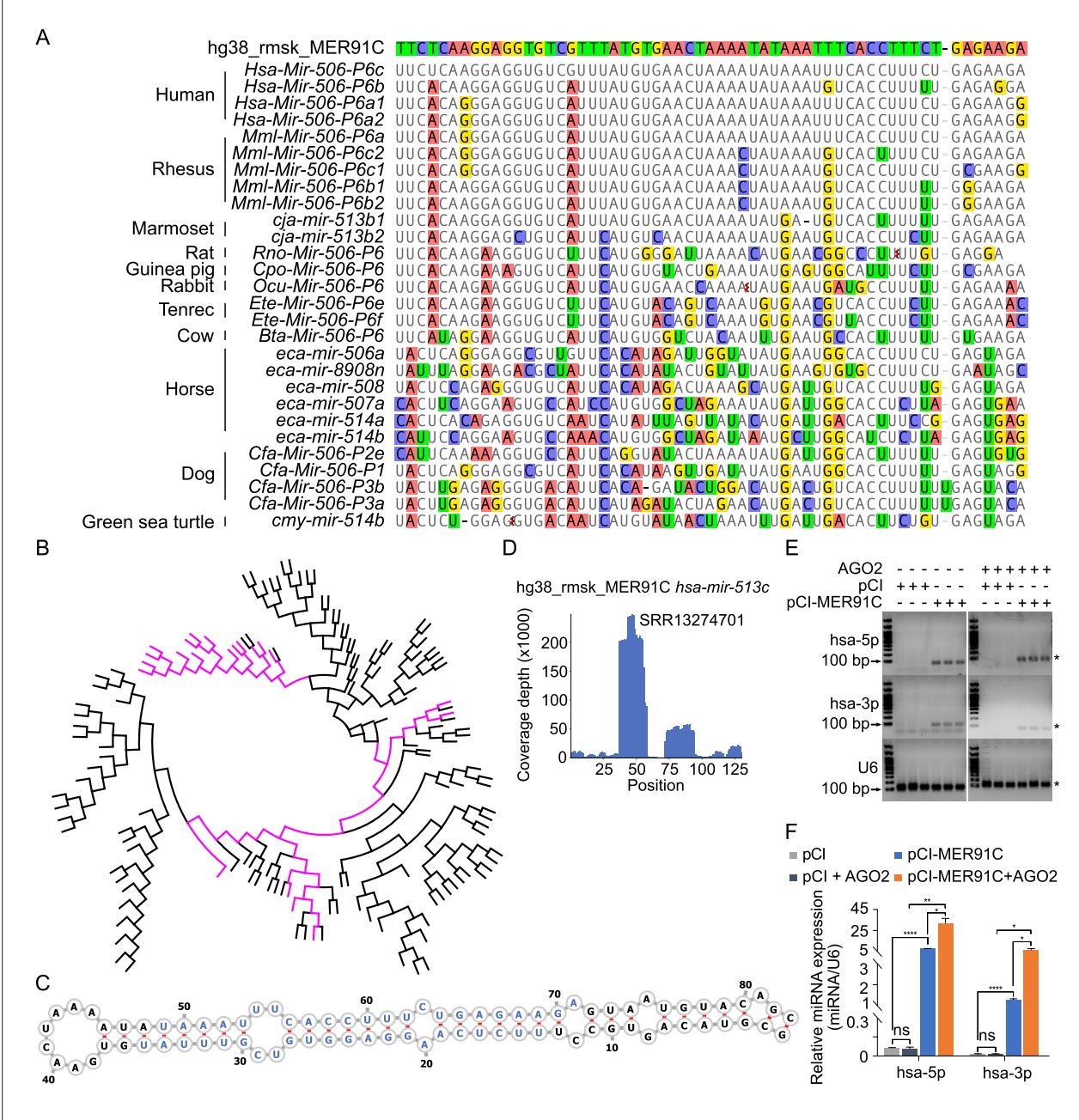

**Figure 2.** Evolutionary history of the X-linked *MIR-506* family. (**A**) Sequences alignment of *FmiRs* from various species using human MER91C DNA transposon as the reference. The first line is the human MER91C DNA transposon, and below are the miRNAs of various species. Mismatched nucleotides are highlighted with various colors. (**B**) A phylogenetic tree of the MER91C DNA transposons and the X-linked *MIR-506* family miRNAs. The MER91C DNA transposons are labeled in purple. (**C**) RNA structure of the MER91C DNA transposon-derived miRNA (human *MIR513C*). (**D**) sRNA-seq reads (lower panel) of the MER91C DNA transposon-derived miRNA (human *MIR513C*). (**E**) Representative gel images showing expression levels of the MER91C DNA transposon-derived miRNA (human *MIR513A1*) in HEK293T cells. n = 3 for each group. The asterisk (*) indicates the expected miRNA size. U6 was used as the loading control. (**F**) qPCR analyses of expression levels of MER91C DNA transposon-derived miRNA (human *MIR513A1*) in HEK293T cells. n = 3 for each group. *, **, and **** indicate adjusted p-value <0.05, 0.01, and 0.0001, respectively. One-way ANOVA was used for statistical analyses.

The online version of this article includes the following source data and figure supplement(s) for figure 2:

**Source data 1.** The original gel images of the MER91C DNA transposon-derived miRNAs from humans expressed in HEK293T cells in *Figure 2E*.

**Source data 2.** The PDF contains *Figure 2E* and the original gel images labeled with the relevant bands.

*Figure 2 continued on next page*

*Figure 2 continued*

**Figure supplement 1.** A phylogram of the X-linked *MIR-506* family.

**Figure supplement 2.** A phylogenetic tree of the MER91C DNA transposons and the X-linked *MIR-506* family miRNAs.

**Figure supplement 3.** X-linked *MIR-506* family is derived from MER91C DNA transposon and expanded *via* LINE retrotransposons.

**Figure supplement 3—source data 1.** The original gel images of the MER91C DNA transposon-derived miRNAs from horses and dogs expressed in HEK293T cells in *Figure 2—figure supplement 3B*.

**Figure supplement 3—source data 2.** The PDF contains *Figure 2—figure supplement 3B* and the original gel images labeled with the relevant bands.

Co-expression of MER91C DNA regions and AGO2 significantly increased the abundance of human MER91C miRNAs, as compared with overexpression of MER91C DNA regions alone (*Figure 2E and F*, *Figure 2—source data 1 and 2*), suggesting that these miRNAs could be loaded onto and protected by AGO2. Taken together, these results indicate that the X-linked *MIR-506* family miRNAs were originally derived from the MER91C DNA transposon.

## X-linked *MIR-506* family miRNAs are predominantly expressed in spermatogenic cells and sperm

Several previous studies have shown that the X-linked *MIR-506* family miRNAs are predominantly expressed in the testis of multiple species (*Wang et al., 2020b*; *Song et al., 2009*; *Zhang et al., 2019*; *Hirano et al., 2014*; *Zhang et al., 2007*; *Koenig et al., 2016*). By analyzing the publicly available small RNA sequencing (sRNA-seq) datasets from multiple species, including humans, rhesus monkeys, mice, rats, rabbits, dogs, and cows (*Fromm et al., 2020*; *Keller et al., 2022*; *Bushel et al., 2020*), we further confirmed that *MIR-506* family miRNAs were indeed highly abundant in the testis, but barely expressed in other organs (*Figure 3—figure supplement 1* and *Supplementary file 3*). To further determine whether these miRNAs are expressed in male germ cells in rodent testes, we conducted sRNA-seq using pachytene spermatocytes, round spermatids, and sperm purified from adult mice (*Figure 4—figure supplement 1A* and *Supplementary file 3*). Consistent with previous data (*Wang et al., 2020b*; *Song et al., 2009*; *Zhang et al., 2019*), these miRNAs were abundantly expressed in spermatogenic cells in murine testes (*Figure 3A*). Approximately 80% of these miRNAs were significantly upregulated (false discovery rate [FDR] < 0.05) when pachytene spermatocytes developed into round spermatids, and ~83.3% were significantly upregulated (FDR < 0.05) when developed into cauda sperm (*Figure 3A*). By analyzing the publicly available sRNA-seq datasets from humans (*Gainetdinov et al., 2018*), marmosets (*Hirano et al., 2014*), and horses (*Li et al., 2019*), we determined the expression patterns of the *MIR-506* family miRNAs in the testes of these species (*Figure 3B–D* and *Supplementary file 3*). The significantly increasing abundance of the *SmiRs* from immature to mature testes in horses (*Li et al., 2019*) supports the elevated expression in haploid male germ cells (round, elongating/elongated spermatids, and sperm) compared to meiotic male germ cells (spermatocytes) (*Figure 3D*). More interestingly, the *SmiRs* and *FmiRs* appear to be differentially expressed in the testes of various species, for example, relative levels of the *SmiRs* were greater than those of the *FmiRs* in mice (*Figure 3A* and *Supplementary file 3*). Both the *SmiRs* and *FmiRs* were highly expressed in horses (*Figure 3D* and *Supplementary file 3*). Levels of the *FmiRs* in marmoset (*Hirano et al., 2014*) and human testes (*Gainetdinov et al., 2018*) were much greater than in those of the *SmiRs* (*Figure 3B and C* and *Supplementary file 3*). Overall, the *MIR-506* family miRNAs are abundant in the testis and predominantly expressed in haploid male germ cells, that is, spermatids and spermatozoa.

## Ablation of X-linked *MIR-506* family miRNAs compromises male fertility due to reduced sperm competitiveness

To define the physiological role of the *MIR-506* family, we sequentially deleted these miRNA genes using CRISPR-Cas9-based genome editing (*Figure 4A*; *Wang et al., 2020b*). We first generated the KO mice lacking either the *Mir883* single cluster (*Mir883* sKO) or the *Mir465* single cluster (*Mir465* sKO) (*Figure 4A*), as these two clusters are the most abundantly expressed in the mouse testes (*Figure 3A* and *Figure 3—figure supplement 1*). No discernible defects were observed, and these KO males developed normally and were fertile (*Figure 4B and C*). On the *Mir883* sKO background,

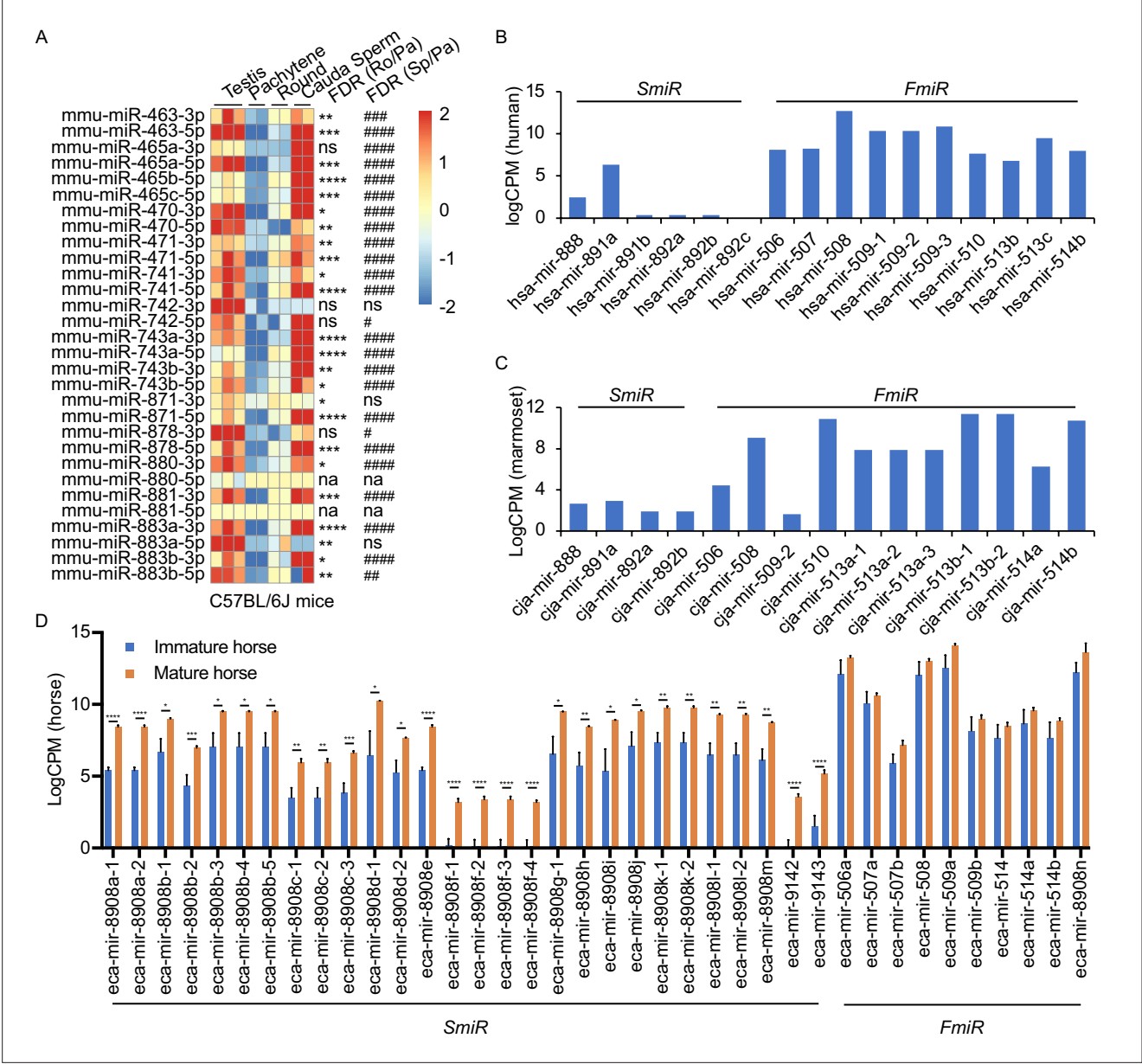

**Figure 3.** Expression profiles of X-linked *MIR-506* family in mammalian testes and male germ cells. (**A**) Heatmaps showing the *MIR-506* family expression in the testis, pachytene spermatocytes, round spermatids, and sperm in mice. Biological triplicates of the testis samples (n = 3) and duplicates of pachytene spermatocytes, round spermatids, and sperm samples isolated from 2 to 4 mice were used for sRNA-seq. *, **, ***, and **** indicate false discovery rate (FDR) <0.05, 0.01, 0.001, and 0.0001, respectively, when comparing round spermatids to pachytene spermatocytes. #, ##, ###, and #### indicate FDR <0.05, 0.01, 0.001, and 0.0001, respectively, when comparing cauda sperm to pachytene spermatocytes. ns and na indicate not significantly and not applicable, respectively. (**B, C**) LogCPM bar graphs showing the *MIR-506* family expression in the testis of humans n = 1 (**B**) and marmosets n = 1 (**C**). (**D**) LogCPM bar graph showing the *MIR-506* family expression in sexually immature and mature horse testes. n = 3. *, **, ***, and **** indicate FDR <0.05, 0.01, 0.001, and 0.0001, respectively.

The online version of this article includes the following figure supplement(s) for figure 3:

**Figure supplement 1.** sRNA-seq of multiple tissues from different species.

we further deleted the *Mir741* cluster, which we termed double KO (dKO) (***Figure 4A***), but no discernible abnormalities were observed in the dKO males either (***Figure 4—figure supplement 1B–E***). On the dKO background, we next deleted either the *Mir465*, termed triple KO (tKO), or the *Mir471* and *Mir470* clusters, termed quadruple KO (quadKO) (***Figure 4A***). Lastly, we ablated the *Mir465* cluster on the quadKO background, named quintuple KO (quinKO) (***Figure 4A***). To reduce potential off-target

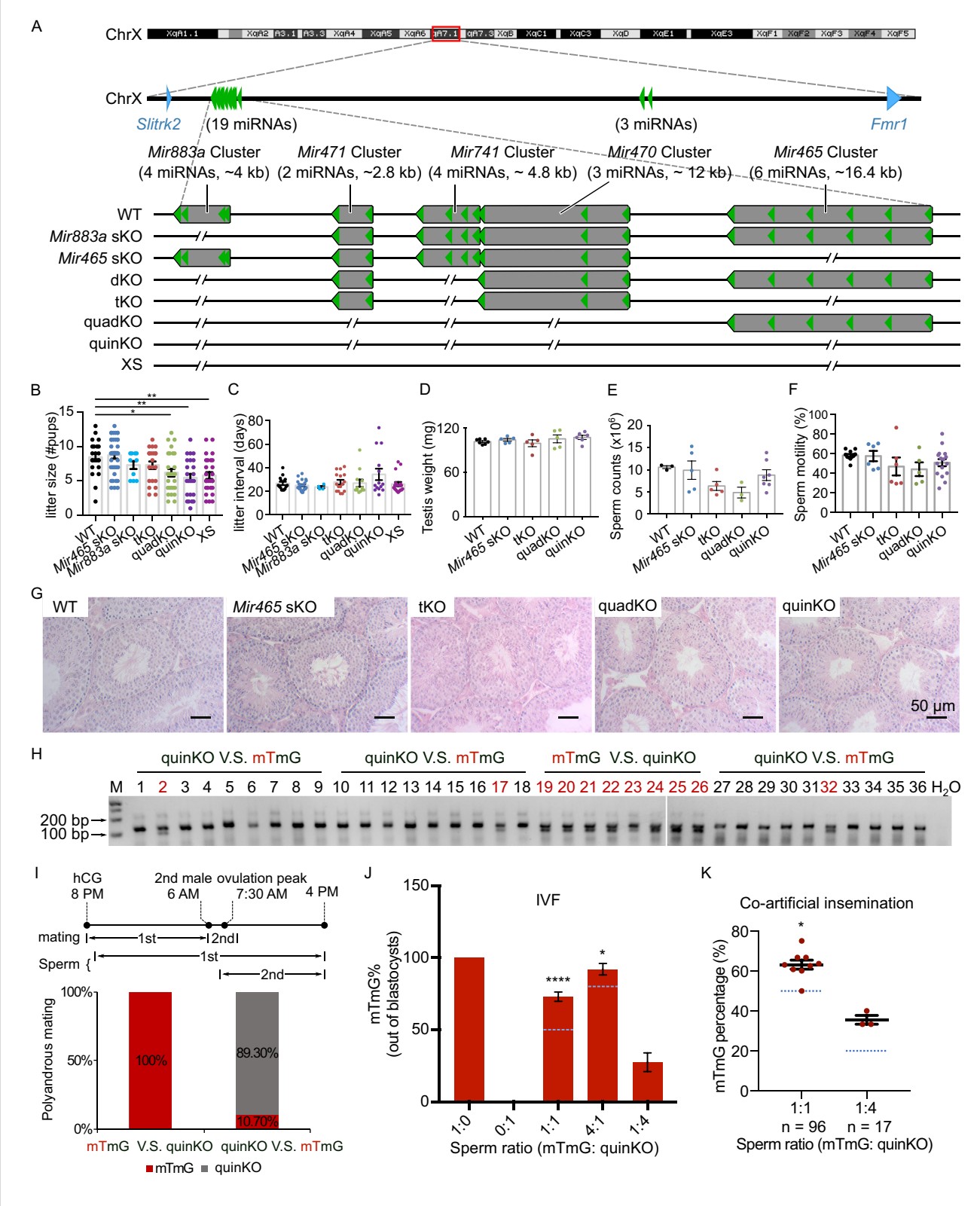

**Figure 4.** Ablation of X-linked *MIR-506* family miRNAs compromised sperm competitiveness and reproductive fitness in male mice. (**A**) Schematics showing the strategy used to generate six lines of KO mice lacking individual or combined miRNA clusters within the *MIR-506* family using CRISPR-Cas9. (**B, C**) Litter size (**B**) and litter interval (**C**) of six *MIR-506* family KO lines, at least 10 litters from three different breeding pairs for each KO line were counted. Dunnett's multiple comparisons test as the post hoc test following one-way ANOVA was used for the statistical analysis. ns, not significant. *

*Figure 4 continued on next page*

*Figure 4 continued*

and ** indicate adjusted p-value <0.05 and 0.01, respectively. (**D–F**) Analyses of testis weight (**D**), sperm counts (**E**), and sperm motility (**F**) in four *MIR-506* family KO lines. n ≥ 3 and Dunnett's multiple comparisons test as the post hoc test following one-way ANOVA was used for the statistical analysis. (**G**) Testicular histology of WT and four *MIR-506* family KO lines showing largely normal spermatogenesis. Scale bars = 50 μm. (**H**) Representative genotyping results of the sequential mating experiments. (**I**) Sequential mating of WT female mice with mTmG and quinKO males. Upper panel, an overview of the polyandrous mating scheme. 'mTmG V.S. quinKO': mTmG male mice mated first; 'quinKO V.S. mTmG': quinKO male mice mated first. (**J**) Percentage of mTmG blastocysts obtained from in vitro fertilization (IVF) using WT MII oocytes and mixed sperm from mTmG (control) and quinKO males at different ratios. Data were based on three independent IVF experiments. The expected ratio was indicated as the blue line. Chi-squared test was used for statistical analyses. * and **** indicate p<0.05 and 0.0001, respectively. (**K**) Percentage of mTmG embryos obtained from co-artificial insemination using different ratios of mTmG and quinKO sperm. Data were based on nine and three independent AI experiments for the 1:1 and 1:4 sperm ratio (mTmG: quinKO), respectively. The expected ratio is indicated as the blue line. Chi-squared test was used for statistical analyses. *p<0.05.

The online version of this article includes the following source data and figure supplement(s) for figure 4:

**Source data 1.** The original gel images of the genotyping results of the sequential mating experiments in *Figure 4H*.

**Source data 2.** The PDF contains *Figure 4H* and the original gel images labeled with the relevant bands.

**Figure supplement 1.** Phenotypes of *MIR-506* family KO mice.

**Figure supplement 1—source data 1.** The original gel images of the T7EI assay on the WT and quinKO mice genomic DNA from tail snips in *Figure 4—figure supplement 1F*.

**Figure supplement 1—source data 2.** The PDF contains *Figure 4—figure supplement 1F* and the original gel images labeled with the relevant bands.

**Figure supplement 1—source data 3.** The original gel images of genotyping the E10 embryos from co-artificial insemination in *Figure 4—figure supplement 1J*.

**Figure supplement 1—source data 4.** The PDF contains *Figure 4—figure supplement 1J* and the original gel images labeled with the relevant bands.

effects due to multiple rounds of CRISPR-Cas9 targeting, we also generated a KO mouse line with only 4 guide RNAs (gRNAs) flanking the *SmiRs* region, named X-linked *SmiRs* KO (XS) (*Figure 4A*). The XS mice were genetically equivalent to the quinKO mice and phenotypically identical to quinKOs (*Figure 4B and C*), suggesting the phenotype observed was not due to the accumulating off-target effects. To further exclude the potential off-target effect, all KO mouse strains were backcrossed with WT C57BL/6J mice for at least five generations before data collection. In addition, T7 endonuclease I (T7EI) assays showed no discernible off-target effects in the quinKO mice (*Figure 4—figure supplement 1F*, and *Figure 4—figure supplement 1—source data 1 and 2*).

While the litter size was still comparable between the tKO and WT control mice, the quadKO, quinKO, and XS males produced significantly smaller litters (~5 vs. ~8 pups/litter) (adjusted p-value<0.05, one-way ANOVA) (*Figure 4B*). Of interest, no significant changes were detected in litter interval, testis weight, or histology in any of the four types of KOs, as compared to WT controls (*Figure 4C–G*). Computer-assisted sperm analyses (CASA) revealed no significant differences in sperm counts and motility parameters among the four types of KOs (*Figure 4E and F*, *Figure 4—figure supplement 1H*). Overall, there appears to be an inverse correlation ($R^2 = 0.9139$, p<0.05, *F*-test) between the number of miRNAs inactivated and the litter size (*Figure 4—figure supplement 1G*). Interestingly, several human studies have correlated the dysregulated *MIR-506* family miRNAs with impaired male fertility due to maturation arrest and oligo-asthenozoospermia (*Supplementary file 4*; *Abu-Halima et al., 2013*; *Heidary et al., 2019*; *Tian et al., 2018*; *Qing et al., 2017*; *Wang et al., 2011*). These data suggest that the *MIR-506* family may play an important role in spermatogenesis and male fertility.

Most of the protein-coding genes that are exclusively or preferentially expressed in the testis with an essential role in spermatogenesis are highly conserved across species (*Xia et al., 2020*). Despite their male germ cell-predominant expression, the *MIR-506* family miRNAs appear to have evolved rapidly to diverge their sequences, suggesting that these miRNAs might control certain 'non-conserved' aspects of spermatogenesis, leading to enhanced sperm competitiveness for male reproductive success. Supporting this hypothesis, previous reports have documented that females of most species throughout the animal kingdoms mate with multiple males before pregnancy, suggesting that sperm competition may serve as a selection mechanism to bias the birth of offspring sired by the males with more competitive sperm (*Dean et al., 2006*; *Firman and Simmons, 2008*). Studies have also shown female rodents in the wild mate with multiple males and produce litters of mixed paternity, and that

pups born to the females following such polyandrous mating display greater survival rates than those produced from females following monandrous mating (*Firman, 2011*). Given that CASA detected no difference in swimming patterns between quinKO and WT sperm (*Figure 4—figure supplement 1H*), we next carried out sperm competition experiments that mimic polyandrous mating in the wild. Since the *MIR-506* family miRNAs are X-linked, the Y sperm from the quinKO mice are genetically indistinguishable from those of WT controls. We, therefore, adopted the mTmG male mice (*Muzumdar et al., 2007*) for sperm competition experiments because the embryos or offspring fathered by the mTmG males can be easily identified based on the constitutively expressed membrane-tagged tomato red (mT) fluorescence and/or PCR genotyping.

We first conducted sequential mating with two mating events ~6–8 hr apart. Interestingly, all of the pups born were fathered by mTmG males (n = 8) when the WT females were mated first with mTmG males and subsequently with the quinKO males. In contrast, when the WT females were mated first with quinKO males and subsequently with mTmG males, ~89% of the pups born were fathered by quinKO males, and the remaining ~11% of pups were from mTmG males (n = 28) (*Figure 4H and I*, *Figure 4—source data 1 and 2*). It is noteworthy that in the sequential mating experiments, the two coituses occurred ~6–8 hr apart due to practical reasons, whereas in the wild, polyandrous mating may take place much faster. To better mimic polyandrous mating in vitro, we mixed the WT and quinKO sperm in different ratios and used the mixed sperm to perform IVF (in vitro fertilization). MII oocytes fertilized by mTmG, quinKO, or a mixture of two types of sperm at three ratios (mTmG:quinKO = 1:1, 4:1, and 1:4) all displayed comparable rates at which fertilized oocytes developed into blastocysts (*Figure 4—figure supplement 1I*). Interestingly, when a 1:1 ratio (mTmG sperm:quinKO sperm) was used, ~73% of the resulting blastocysts were derived from mTmG sperm, whereas the remaining ~27% were from quinKO sperm (n = 179) ($p < 0.0001$, chi-squared test) (*Figure 4J*). When a 4:1 sperm ratio (mTmG: quinKO) was used, ~92% of the blastocysts were from mTmG sperm and only 8% were from quinKO sperm (n = 170) ($p < 0.05$, chi-squared test) (*Figure 4J*). In contrast, when a 1:4 sperm ratio (mTmG:quinKO) was used, blastocysts derived from mTmG and quinKO sperm represented ~28% and ~72% of the total, respectively (n = 135) (*Figure 4J*). We also performed co-artificial insemination (AI) using mTmG and quinKO sperm. When a 1:1 sperm ratio (mTmG:quinKO) was used, ~62.5% of the embryos (n = 96) were derived from mTmG sperm ($p < 0.05$, chi-squared test) (*Figure 4K*). When a 1:4 ratio was used, ~35.3% of the embryos (n = 17) were from the mTmG mice (*Figure 4K* and *Figure 4—figure supplement 1J*, *Figure 4—figure supplement 1—source data 3 and 4*). Together, these results indicate that the quinKO sperm are less competitive than the control mTmG sperm both in vivo and in vitro. Previous studies suggest that sperm aggregation and midpiece size might be involved in sperm competitiveness (*Fisher and Hoekstra, 2010*; *Fisher et al., 2016*), but no changes in these two parameters were observed in the quinKO sperm (*Figure 4—figure supplement 1K and L*). Although the blastocyst rate (out of two-cell embryos) of quinKO was comparable to that of the mTmG mice, the two-cell rates (out of zygotes) were significantly reduced ($p < 0.05$, paired *t*-test) in the quinKO mice (~39%, n = 67) when compared to the mTmG mice (~88%, n = 58) (*Figure 4—figure supplement 1M*), implying that the quinKO sperm is indeed less efficient in fertilizing eggs and/or supporting early embryonic development, especially the first cleavage of the zygotes.

## X-linked *MIR-506* family miRNAs mostly target the genes involved in spermatogenesis and embryonic development and compensate for each other

To identify the target genes of these X-linked *MIR-506* family miRNAs, we performed RNA-seq analyses using testis samples from the five types of KOs (*Mir465* sKO, dKO, tKO, quadKO, and quinKO) (*Figure 5—figure supplement 1A* and *Supplementary file 5*). Comparisons between the KO and WT testes revealed thousands of differentially expressed genes (DEGs) (fold change $\geq$ 2, FDR < 0.05, *Figure 5—figure supplement 1A* and *Supplementary file 5*). The DEGs identified were then compared with the predicted *MIR-506* target genes using four different databases, including TargetScan (*Agarwal et al., 2015*), microrna.org (*Betel et al., 2010*), miRWalk (*Dweep and Gretz, 2015*), and mirDB (*Chen and Wang, 2020*), to predict the differentially expressed targets (DETs) of the *MIR-506* family miRNAs (*Figure 5—figure supplement 1A* and *Supplementary file 5*). We obtained 2692, 2028, 1973, 3405, and 1106 DETs from *Mir465* sKO, dKO, tKO, quadKO, and quinKO testes, respectively. GO terms of DETs from each KO testis revealed that the DETs were mostly involved in

embryonic development, response to stimulus, centrosome cycle, epithelium morphogenesis, organelle organization, cell projection, RNA metabolic process, and DNA repair (*Figure 5—figure supplement 1B*). The 431 DETs identified to be shared across all five KO testes were also enriched in similar pathways (*Figure 5A and B*). Several genes, including *Crisp1*, *Egr1*, and *Trpv4*, were selected for validation using qPCR, Western blots and luciferase-based reporter assays. Consistent with the RNA-seq data, qPCR showed that *Crisp1*, *Egr1*, and *Trpv4* were significantly downregulated in the quinKO testes (*Figure 5—figure supplement 1C*). CRISP1 is enriched in the sperm principal piece and head (*Figure 5—figure supplement 1D*). Western blots also confirmed that *CRISP1* is downregulated in the quinKO testis when compared to the WT testis (*Figure 5—figure supplement 1E*, *Figure 5—figure supplement 1—source data 1 and 2*). Luciferase assays further confirmed that *Egr1* and *Crisp1* are targets of the *MIR-506* family members (*Figure 5—figure supplement 1F and G*). *Egr1* 3′UTR luciferase activity was upregulated by miR-465c, while downregulated by miR-743b (*Figure 5—figure supplement 1F*). miR-465a, miR-465c, miR-470, miR-741, and miR-743a upregulated *Crisp1* 3′UTR luciferase activity, while miR-743b exerted the opposite effect (*Figure 5—figure supplement 1G*). Of interest, KO of *Crisp1* in mice or inhibition of *CRISP1* in human sperm appears to phenocopy the quinKO mice (*Da Ros et al., 2008*; *Maldera et al., 2014*). Specifically, sperm motility in the *Crisp1* KO mice is comparable to that in WT mice, but their ability to penetrate the eggs was reduced in the *Crisp1* KO mice (*Da Ros et al., 2008*); a similar effect was also observed in human sperm treated with anti-hCRISP1 antibody (*Maldera et al., 2014*).

The inverse correlation between the number of miRNAs inactivated and the severity of the phenotype strongly hints that these miRNAs compensate for each other (*Figure 4B* and *Figure 4—figure supplement 1G*). To test this hypothesis, we performed sRNA-seq on four KO (*Mir465* sKO, tKO, quadKO, and quinKO) testes. The sRNA-seq data showed that these miRNAs were no longer expressed in the corresponding KOs, confirming the successful deletion of these miRNAs in these KOs (*Figure 5C* and *Supplementary file 6*). Interestingly, in *Mir465* sKO testes, miR-201, miR-547, miR-470, miR-471, miR-742, miR-871, miR-881, miR-883a, and miR-883b were all significantly upregulated (FDR <0.05). Similarly, miR-201, miR-547, miR-470, miR-471, miR-871, and miR-883b were all significantly upregulated in the tKO testes (FDR < 0.05); miR-201 and miR-547 were all significantly upregulated in the quadKO and the quinKO testes (FDR < 0.05) (*Figure 5C* and *Supplementary file 6*). These results support the notion that genetic compensation exists among the X-linked *MIR-506* family miRNAs.

## Rapid evolution of the *MIR-506* family is not driven by the increased complexity of 3′UTRs of the conserved targets but rather adaptive to targeting more genes

To minimize false positives, the DETs in mice were selected using the following criteria: (1) dysregulated by fold change ≥ 2 and FDR < 0.05. (2) Falling within the predicted targets. (3) Intersected with at least two different KO mouse samples. Using the 3043 DETs identified in mice as a reference, we searched the predicted targets of the *MIR-506* family miRNAs in rats and humans to determine if these target genes were shared across species (*Figure 5A*). While 2098 (~69%) target genes were shared among all three species, 2510 (~82%) were common to both humans and mice, and 2202 (~72%) were shared between mice and rats (*Figure 6A* and *Supplementary file 7*). To test the accuracy of the predicted targets, we selected several genes in humans and performed luciferase assays using human *MIR-506* family miRNAs and their corresponding target genes (*Figure 6—figure supplement 1A and B*). Among these targets, *CRISP1* and *FMR1* were shared among humans, mice, and rats, and confirmed to be targeted by the *MIR-506* family miRNAs in mice (*Figure 5—figure supplement 1C, E, and G*; *Guo et al., 2022*; *Wang et al., 2020b*). Luciferase assays also confirmed that human *CRISP1* (*hCRISP1*) and *FMR1* (*hFMR1*) were targets of the *MIR-506* family, and miR-510 and miR-513b both could activate *hCRISP1* 3′UTR luciferase activity (*Figure 6—figure supplement 1A*), whereas miR-509-1, miR-509-2, miR-509-3, miR-513b, miR-514a, and miR-514b could enhance *hFMR1* 3′UTR luciferase activity (*Figure 6—figure supplement 1B*). These results confirmed the accuracy of our predicted targets in humans.

We considered two likely explanations for the paradox where the majority of their target genes were shared across species despite the rapid evolution of the *MIR-506* family miRNAs: (1) the 3′UTR sequences in extant target genes became increasingly divergent during evolution such that the

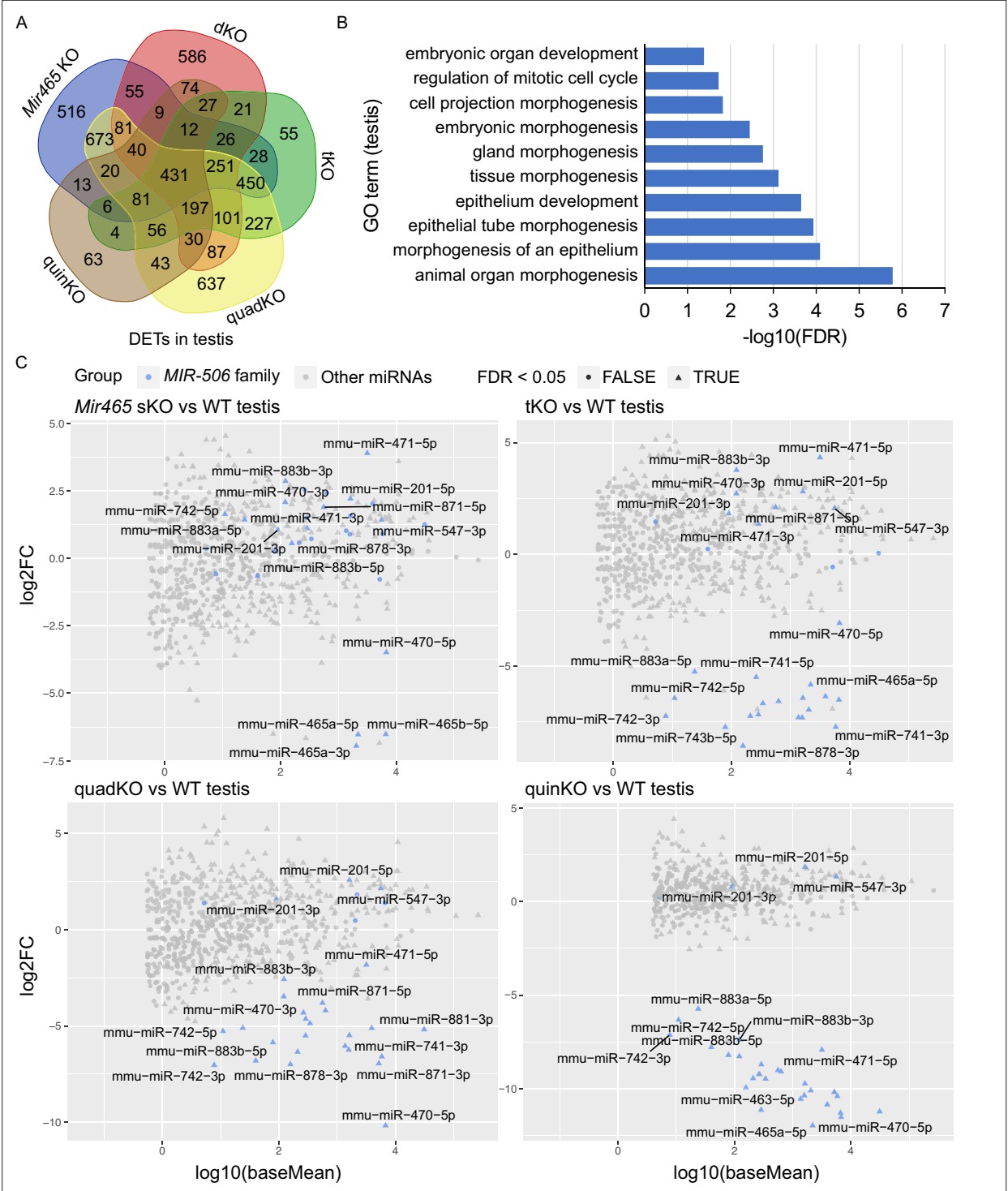

**Figure 5.** Target genes and genetic compensation of the X-linked *MIR-506* family miRNAs. (**A**) Intersections of the differentially expressed targets (DETs) among different KO testes. (**B**) GO term enrichment analyses of the 431 DETs shared among the four different *MIR-506* family KO testes. (**C**) MA plots showing the expression levels of the *MIR-506* family miRNAs in WT, sKO, tKO, quadKO, and quinKO testes. Three biological replicates (n = 3) were used for sRNA-seq analyses.

The online version of this article includes the following source data and figure supplement(s) for figure 5:

**Figure supplement 1.** Dysregulated targets in the X-linked *MIR-506* family KO testes.

**Figure supplement 1—source data 1.** The original Western blot of *CRISP1* in WT and quinKO testis samples in *Figure 5—figure supplement 1E*.

*Figure 5 continued on next page*

*Figure 5 continued*

**Figure supplement 1—source data 2.** The PDF contains *Figure 5—figure supplement 1E* and the original WB images labeled with the relevant bands.

---

*MIR-506* family miRNAs had to adapt to maintain their ability to bind these 3′UTRs; or (2) that the *MIR-506* family miRNAs evolved rapidly in a manner that allowed them to target mRNAs encoded by additional genes involved in spermatogenesis. To distinguish the two possibilities, we first compared the extent of similarities among the 3′UTR sequences of the 2510 shared target genes between humans and mice (*Figure 6A* and *Supplementary file 7*). We adopted the PhyloP scores to measure the evolutionary conservation at individual nucleotide sites in the 3′UTRs of the shared target genes. The overall conservation appeared to be greater in the regions targeted by *MIR-506* family miRNAs than in the non-target regions in the 3′UTRs of the shared target genes in both mice and humans (*Figure 6B and C*) with a few exceptions (*Figure 6D and E*) ($p<0.05$, *t*-test). These data suggest that the regions targeted by the X-linked *MIR-506* family miRNAs are under relatively stronger purifying, rather than adaptive, selection. We then tested the second hypothesis that the rapid evolution of the *MIR-506* family resulted in more extant mRNAs being targeted by these miRNAs. We first compared the average target numbers of each *MIR-506* family miRNA between humans and mice using the 2510 shared targets between predicted targets in humans and the dysregulated targets in mice (*Figure 6A and F*).

Among these shared targets, the human *MIR-506* family members could target ~1268 unique transcripts per miRNA, whereas the murine *MIR-506* family members could only target ~1068 ($p<0.05$, *t*-test) (*Figure 6F*), indicating that the *MIR-506* family miRNAs target more genes in humans than in mice. Furthermore, we analyzed the number of all potential targets of the *MIR-506* family miRNAs predicted by the aforementioned four algorithms among humans, mice, and rats. The total number of targets for all the X-linked *MIR-506* family miRNAs among different species did not show significant enrichment in humans (*Figure 6—figure supplement 1C*), suggesting the sheer number of target genes does not increase in humans. We then compared the number of target genes per miRNA. When comparing the number of target genes per miRNA for all the miRNAs (baseline) between humans and mice, we found that on a per miRNA basis, human miRNAs have more targets than murine miRNAs ($p<0.05$, *t*-test) (*Figure 6—figure supplement 1D*), consistent with higher biological complexity in humans. This became even more obvious for the X-linked *MIR-506* family ($p<0.05$, *t*-test) (*Figure 6— figure supplement 1D*). In humans, the X-linked *MIR-506* family, on a per miRNA basis, targets a significantly greater number of genes than the average of all miRNAs combined ($p<0.05$, *t*-test) (*Figure 6—figure supplement 1D*). In contrast, in mice, we observed no significant difference in the number of targets per miRNA between X-linked miRNAs and all of the mouse miRNAs combined (mouse baseline) (*Figure 6—figure supplement 1D*). These results suggest that although the sheer number of target genes remains the same between humans and mice, the human X-linked *MIR-506* family targets a greater number of genes than the murine counterpart on a per miRNA basis. We also investigated the number of *MIR-506* family miRNA targeting sites within the individual target genes in both humans and mice, but no significant differences were found between humans and mice (*Figure 6G*). To determine whether increased target sites in humans were due to the expansion of the MER91C DNA transposon, we analyzed the MER91C DNA transposon-containing transcripts and associated them with our DETs. Of interest, 28 human and 3 mouse mRNAs possess 3′UTRs containing MER91C DNA sequences, and only 3 and 0 out of those 28 and 3 genes belonged to DETs in humans and mice, respectively (*Figure 6—figure supplement 1E*), suggesting a minimal effect of MER91C DNA transposon expansion on the number of target sites. Taken together, these results suggest the human X-linked *MIR-506* family has been subjected to additional selective pressure, causing them to exert additional regulatory functions by targeting more mRNAs expressed during spermatogenesis (*Figure 6H*).

## Discussion

Successful reproduction is pivotal for the perpetuation of species, and sperm are constantly facing selective pressures (*Morrow, 2004*). To enhance their chance to fertilize eggs, sperm need to adapt accordingly, and miRNAs-mediated regulation of gene expression in spermatogenesis provides a

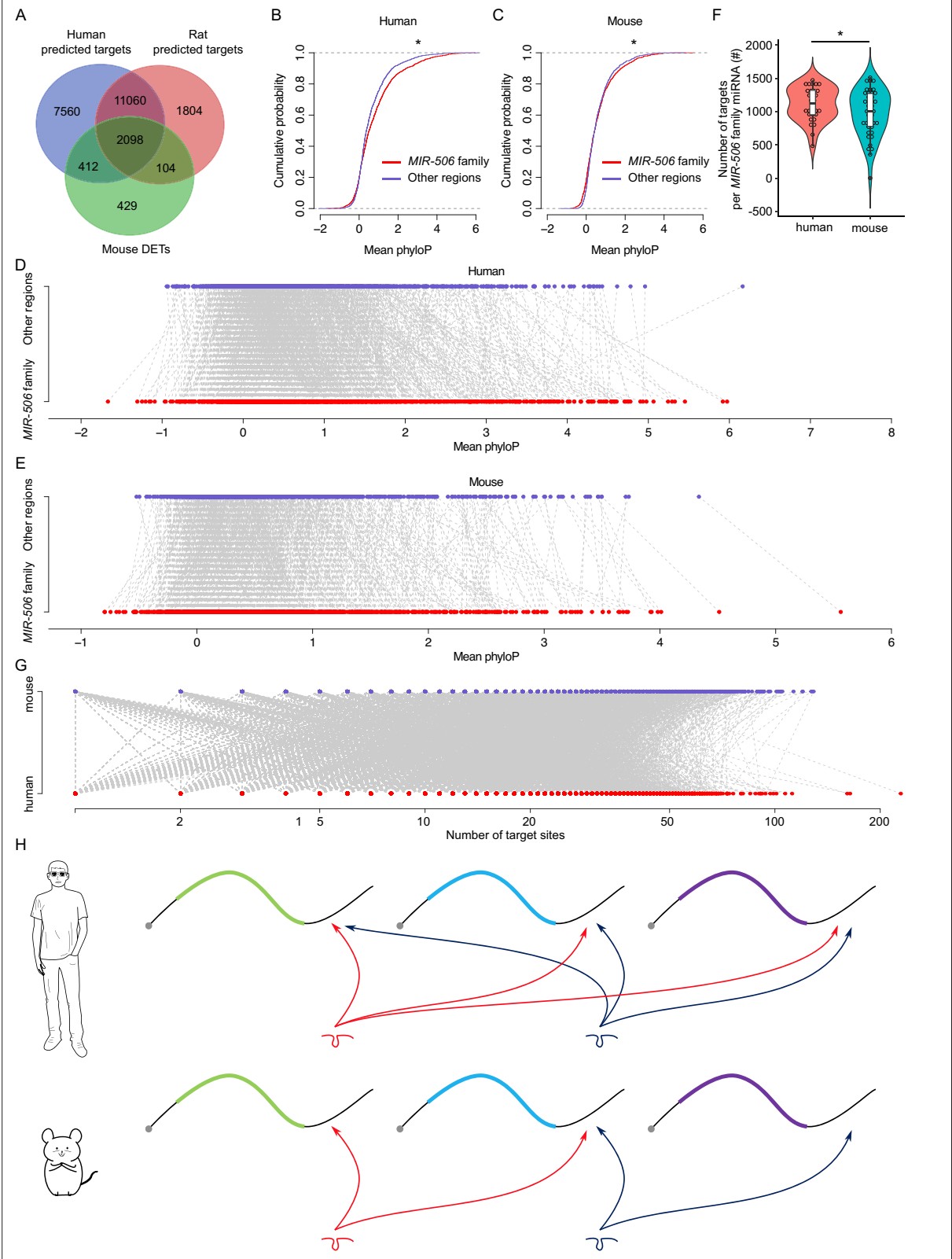

**Figure 6.** Rapid evolution of the X-linked *MIR-506* family miRNAs correlates with increased complexity of genetic networks that regulate spermatogenesis across mammalian species. (**A**) Overlap between the dysregulated targets in mice and the predicted targets in humans and rats. (**B**) Comparison of the cumulative distribution between the *MIR-506* family targeting sites and the other regions in humans. *p<0.05; *t*-test was used for statistical analyses. (**C**) Comparison of the cumulative distribution between the *MIR-506* family targeting sites and the other regions in mice. *p<0.05;

*Figure 6 continued on next page*

*Figure 6 continued*

t-test was used for statistical analyses. (**D**) Paired comparison of the PhyloP score between the *MIR-506* family targeting sites and the other regions in humans. (**E**) Paired comparison of the PhyloP score between the *MIR-506* family targeting sites and the other regions in mice. (**F**) Comparison of the number of the targets per miRNA for the X-linked *MIR-506* family in mice and humans. *p<0.05; t-test was used for statistical analyses. (**G**) The number of target sites within individual target mRNAs in both humans and mice. (**H**) Schematics show that human *MIR-506* family miRNAs have more targets relative to those of mice during evolution.

The online version of this article includes the following figure supplement(s) for figure 6:

**Figure supplement 1.** Dysregulated targets are shared across humans, mice, and rats.

rapidly adaptable mechanism toward this end. Although miRNAs were initially believed to be evolutionarily conserved, the number of non-conserved miRNAs has been steadily increasing (*Piriyapongsa et al., 2007*). Among the non-conserved miRNAs, many are derived from TEs, suggesting that TEs may serve as a major source of miRNA sequences (*Piriyapongsa et al., 2007*). The delayed recognition of TE-derived miRNAs, in part, results from the fact that repetitive sequences were usually excluded during the computational annotation of miRNAs. In theory, TEs can serve as a good donor of miRNA sequences for the following reasons: (1) TEs are ubiquitous and abundant in the genome and are known to contribute to the regulatory elements of the coding genes, for example, UTRs (*Jordan et al., 2003*). As one of the regulatory factors that target mainly the 3′UTRs, TE-derived miRNAs can regulate a larger number of mRNAs with multiple miRNA-targeting sites. (2) TEs are among the most rapidly evolving sequences in the genome (*González and Petrov, 2012*) and thus, can continuously produce species/lineage-specific miRNA genes to diversify their regulatory effects. Consistent with these notions, the present study provides evidence supporting that the *MIR-506* family miRNAs originated from the MER91C DNA transposons. Of more interest, the *MIR-506* family miRNAs, despite their rapid evolution, are all expressed in spermatogenic cells in the testis and sperm, supporting a lineage-specific functional diversification of TE-derived miRNAs. In fact, the *MIR-506* family miRNAs were among the first reported TE-derived miRNAs because of their location in a small region of the X chromosome and their confined, abundant expression in the testis (*Song et al., 2009*). RNAs are much less abundant in sperm than in somatic or spermatogenic cells (~1/100) (*Wang et al., 2023*). Sperm-borne small RNAs represent a small fraction of total small RNAs expressed in their precursor spermatogenic cells, including spermatocytes and spermatids (*Wang et al., 2023*). Therefore, when the same amount of total/small RNAs are used for quantitative analyses, sperm-borne small RNAs (e.g., *MIR-506* family miRNAs) would be proportionally enriched in sperm compared to other spermatogenic cells.

It has been demonstrated that under certain circumstances genes that evolve under sexual conflicts tend to move to the X chromosome, especially when they are male-beneficial, female-deleterious, and act recessively (*Rice, 1984*; *Gibson et al., 2002*). The X chromosome is enriched with genes associated with male reproduction (*Wang et al., 2020b*; *Song et al., 2009*; *Wang et al., 2001*). The rapid evolution of the X-linked *MIR-506* family strongly suggests that these miRNA genes were under selection to expand and diversify their regulatory effects on spermatogenesis. Indeed, our data strongly suggest that targeting new mRNAs was likely the driving force for the rapid evolution of the *MIR-506* family of miRNAs. However, expansion and sequence divergence of the X-linked *MIR-506* family may simply reflect natural drifting without functional significance, similar to some of the pachytene piRNA clusters (*Özata et al., 2020*). We argue that the neutral drifting theory may not be true to the *MIR-506* family for the following reasons: (1) despite highly divergent overall sequences of the *MIR-506* family, some miRNAs share the same seed regions across multiple species, suggesting that these regions may undergo strong selections. (2) The *MIR-506* family miRNAs, especially the *FmiRs*, are highly conserved in modern humans, implying a strong selection of these miRNAs. (3) Knockout of the *MIR-506* family (either quinKO or XS) results in male subfertility, reflecting a biological function. (4) The quinKO sperm are less competitive than the WT sperm both in vivo and in vitro. (5) Several human studies have linked the dysregulation of the *MIR-506* family with male infertility/subfertility (*Abu-Halima et al., 2013*; *Heidary et al., 2019*; *Tian et al., 2018*; *Qing et al., 2017*; *Wang et al., 2011*). Similarly, one study in *Drosophila* also showed that the rapidly evolving testis-restricted miRNAs underwent adaptive evolution rather than neutral drifting (*Mohammed et al., 2014*).

Since TEs are abundant in UTRs, TE-derived miRNAs can target a much greater number of mRNAs than those derived from distinct non-repetitive genomic loci (*Piriyapongsa et al., 2007*). Indeed,

thousands of the dysregulated genes detected in the *Mir465* sKO, dKO, tKO, quadKO, and quinKO testes are involved in multiple pathways of spermatogenesis. By analyzing the sequence divergence, we noticed that the most common sequence substitutions among all of the *MIR-506* family miRNA sequences were U-to-C and A-to-G, which were likely mediated by ADARs (adenosine deaminases acting on RNA) that can change A to I (which is functionally equivalent to G) (**Nishikura, 2016**). Interestingly, ~90% of the A-to-I editing appears to have occurred in Alu elements (belonging to the SINE family), and some of the edits occurred in miRNAs (**Nishikura, 2016**). Since G and U can form the so-called G-U wobble base pair (**Varani and McClain, 2000**), those U-to-C or A-to-G substitutions can, in theory, target similar sequences and exert regulatory functions (**Doench and Sharp, 2004**), suggesting that the evolving miRNA sequences could target not only the original sequences but also new sites with similar sequences. Consistent with this notion, the predicted target genes of the *MIR-506* family in mice can also be found in rats and humans, suggesting the target genes are shared across species despite the quick divergence of the miRNA sequences across species. This is also supported by our data showing that the binding sites for the *MIR-506* family of miRNAs are more conserved than the surrounding, non-targeting regions in the 3′UTRs of the predicted target mRNAs. Furthermore, seed sequences among some *MIR-506* family miRNAs remain the same despite the high divergence of these miRNAs, and these conserved seed sequences appear to be present in the dominant mature miRNAs. Thus, the seed region of these miRNAs appears to have undergone strong selection. Supporting this notion, previous studies have shown correlations between miRNA expression and the evolution of miRNAs and target sites (**Simkin et al., 2020**; **Meunier et al., 2013**). In general, miRNAs repress their target gene expression. However, numerous studies have also shown that some miRNAs, such as human miR-369–3, Let-7, and miR-373, mouse miR-34/449 and the *MIR-506* family, and the synthetic miRNA miRcxcr4, activate gene expression both in vitro (**Vasudevan et al., 2007**; **Place et al., 2008**) and in vivo (**Guo et al., 2022**; **Wang et al., 2020b**; **Yuan et al., 2019**; **Yuan et al., 2021**). Earlier reports have shown that these miRNAs can upregulate their target gene expression, either by recruiting *FXR1*, targeting promoters, or sequestering RNA subcellular locations (**Guo et al., 2022**; **Vasudevan et al., 2007**; **Place et al., 2008**). Of interest, miRNAs with the same seed sequences may exert divergent functions. For example, the mature miR-465a, miR-465b, and miR-465c only have a few mismatches outside of the seed region, but only miR-465c exerts functional activation of *Egr1*. A similar effect has also been reported in the miR-465 cluster on the *Alkbh1* 3′UTR activity (**Wang et al., 2022**). Similarly, despite the same seed sequences in the miR-465 or miR-743 cluster, miR-465a and miR-465c have differential activating effects on the 3′UTR of *Crisp1*, and miR-743a and miR-743b exert opposite effects on the *Crisp1* 3′UTR, further confirming their functional divergence. Therefore, the sequences outside of the miRNA seed region may play an important role in their functions, which have also been observed in *C. elegans* and human HEK293 cells (**Wang et al., 2022**; **Broughton et al., 2016**; **Helwak et al., 2013**). To unequivocally demonstrate the physiological role of miRNAs, it would be ideal to delete not only the miRNAs but also their binding sites in their target transcripts in vivo. A few previous studies have established the miRNA: target relationship by deleting the miRNA-binding sites in target transcripts in *C. elegans*, *Drosophila*, and cell lines (**Pinzón et al., 2017**; **Ecsedi et al., 2015**; **Garaulet et al., 2020**), but similar studies have not been reported in mice or humans. The strategy may work in mRNAs with 3′UTRs containing only one or two miRNA-binding sites, but for more complex 3′UTRs of mRNAs in mice and humans that often contain multiple binding sites for the same or different miRNAs, deletion of one miRNA binding site may not cause any discernible effects as the loss of function can easily be compensated by other miRNAs. Nevertheless, by deleting all five highly expressed clusters of the *MIR-506* family one by one, we were able to overcome the compensatory effects among the family members/clusters and successfully revealed the physiological role of this miRNA family. Based on small RNA-seq, some *FmiRs*, for example, *miR-201* and *miR-547*, were upregulated in the *SmiRs* KO mice, suggesting that this small cluster may act in concert with the other five clusters and thus, worth further investigation.

It is well known that mice in the wild are promiscuous, and one female often mates with multiple males sequentially, giving rise to polyandrous litters derived from sperm from more than one sire (**Dean et al., 2006**; **Firman and Simmons, 2008**). Polyandrous mating establishes a situation where sperm from multiple males coexist in the female reproductive tract, with the most competitive ones fertilizing eggs and producing offspring (**Parker, 1970**; **Gomendio et al., 2006**). Therefore, a male that may be fertile in the monandrous mating scheme may rarely sire offspring in a polyandrous mating scenario,

rendering this male functionally 'sub-fertile' or vene 'infertile'. Therefore, sperm competitiveness reflects the general reproductive fitness of the male (*Gomendio et al., 2006*). Although the quinKO males tend to produce smaller litters under the monandrous mating scheme, their sperm counts, sperm motility, and morphology are indistinguishable from those of WT sperm. This is not surprising given that miRNAs of the *MIR-506* family most likely function to control certain non-essential aspects of spermatogenesis. Sperm can be subject to competition at multiple steps during fertilization, including their migration through the female reproductive tract (cervix, uterine cavity, and oviduct), binding the cumulus-oocyte complexes, penetration of zona pellucida, etc. Therefore, IVF may not be ideal for evaluating sperm competition in the real world as it bypasses several key sites where sperm competition likely takes place. AI may represent a better way to assess sperm competition than IVF, but it is probably less desirable than polyandrous mating for the following reasons: first, in the wild, sperm from two males rarely, if not never, enter the female reproductive tract simultaneously. We had tried to place two males into the cage with one female, but the two males ended up fighting, and the submissive one never mated. Second, sperm are delivered directly into the uterus or oviduct during AI (*Nagy et al., 2003*; *Stone et al., 2015*), thus bypassing the potential sites for sperm competition (e.g., cervix and uterine cavity). Although our breeding scheme also involves sperm competition, by shortening the time between the two mating events in a laboratory setting, the sequential mating method reported here may be further improved to better mimic the natural polyandrous mating in the future. Moreover, future analyses of the quinKO sperm may help identify biochemical or molecular biomarkers for sperm competitiveness.

In summary, our data suggest that the *MIR-506* family miRNAs are derived from the MER91C DNA transposon. These miRNAs share many of their targets and can compensate for each other's absence, and they work jointly through regulating their target genes in spermatogenesis to ensure sperm competitiveness and male reproductive fitness.

## Materials and methods

### Animal care and use

All mice used in this study were on 2- to 3-month-old adult C57BL/6J background (strain # 000664, The Jackson Laboratory, RRID:IMSR_JAX:000664) and housed in a temperature- and humidity-controlled, specific pathogen-free facility under a light-dark cycle (12:12 light-dark) with food and water ad libitum. Animal use protocol was approved by the Institutional Animal Care and Use Committees (IACUC) of the University of Nevada, Reno (protocol: 00494) and The Lundquist Institute at Harbor-UCLA (protocol: 32132-03), and is following the 'Guide for the Care and Use of Experimental Animals' established by the National Institutes of Health (1996, revised 2011).

### Generation of the knockout mice

The single, double, triple, quadruple, and quintuple *MIR-506* family miRNAs KO mice were generated as previously described (*Wang et al., 2020b*). Briefly, Cas9 mRNA (200 ng/µl) and gRNAs flanking the *MIR-506* family subclusters (100 ng/µl) were mixed and injected into the cytoplasm of zygotes in the M2 medium. After injection, all embryos were cultured for 1 hr in KSOM + AA medium (Cat# MR-121-D, Millipore) at 37°C under 5% $CO_2$ in the air before being transferred into 7- to 10-week-old female CD1 recipients.

The *Mir883* sKO or the *Mir465* sKO mice were first generated. After at least two rounds of back-crossing with C57BL/6J mice, the *Mir741* cluster was knocked out on the *Mir883* sKO background, which was termed dKO. After at least two rounds of backcrossing the dKO with C57BL/6J mice, the *Mir465* cluster and *Mir471* & *Mir470* clusters were further deleted, which was termed tKO and quadKO, respectively. Lastly, the *Mir465* cluster was ablated on the quadKO background, which was named quinKO. The XS mice were generated by using only four gRNAs flanking the *SmiRs* region on the C57BL/6J background. All KO mice were backcrossed with the C57BL/6J mice for at least five generations before collecting data. WT and KO mice were selected randomly for all experiments.

### Sequential polyandrous mating

Sequential polyandrous mating was carried out based on the ovulation time point (10–13 hr after hCG) as previously described (*Nagy et al., 2003*). Adult (8–12 wk of age) C57BL/6J females were

injected (i.p.) with 7 IU PMSG at 8 p.m., followed by 7 IU hCG 48 hr later. After hCG, the first male mouse was put into the cage of one female from 8 p.m. to 6 a.m. the next day. The first plug was marked with a marker pen. A second male mouse was then introduced into the cage of the plugged female, which was checked every 30–40 min to identify a new plug (non-marked). Females that were plugged twice were kept for producing pups for paternity analyses.

## In vitro fertilization (IVF)

Adult (8–12 wk) C57BL/6J female mice were first treated with 7 IU pregnant mare serum gonado-tropin (PMSG, Cat# HOR-272, Prospecbio) through i.p. injection followed by i.p. injection of 7 IU hCG 48 hr later. Oocytes were collected from the ampulla ~14 hr after the hCG (Cat# HOR-250, Prospecbio) treatment, and the cumulus cells surrounding oocytes were removed by treatment with bovine testicular hyaluronidase (1.5 mg/ml; Cat# H3506, Sigma) in M2 (Cat# MR-015-D, Millipore) at 37°C for 2 min. The cumulus-free oocytes were washed and kept in equilibrated HTF (Cat# MR-070-D, Millipore) at a density of 20–30 oocytes per 60 µl HTF at 37°C in an incubator with air containing 5% $CO_2$ prior to IVF. Cauda epididymal sperm were collected in 100 µl of equilibrated HTF medium, allowing spermatozoa to capacitate for ~30 min at 37°C in an incubator containing 5% $CO_2$ air. After capacitation, spermatozoa (2 µl) were diluted by tenfold and subjected to CASA using the Sperm Analyzer Mouse Traxx (Hamilton-Thorne). Based on the sperm concentration, an aliquot of $2.5 \times 10^8$ spermatozoa was added into each HTF-oocytes drop (~60 µl) for IVF. Then, ~4 hr later, zygotes were washed and cultured in KSOM + AA (Cat# MR-121-D, Millipore) until the blastocyst stage at 37°C in an incubator with air containing 5% $CO_2$. The two-cell embryos were counted 24–26 hr after IVF, and blastocysts were counted and analyzed under a fluorescence micro-scope 70–72 hr after IVF.

## Artificial insemination (AI)

At least 2-month-old female CD1 or C57BL/6J mice were administered with 2.5 IU of PMSG (Cat# HOR-272, Prospecbio) at 5:30 p.m. 3 d before artificial insemination, followed by 2.5 IU of hCG (Cat# HOR-250, Prospecbio) at 5:00 p.m. 1 d prior to AI. The next morning at 8:00 a.m., ~2-month-old mTmG and quinKO male mice were sacrificed, the cauda epididymis was dissected, and fat tissue and blood were removed before placing the cauda epididymis into 500 µl or 150 µl of EmbryoMax Human Tubal Fluid (HTF) (1×) (Cat# MR-070-D, MilliporeSigma) containing 4 mg/ml BSA (Cat# 12659-250GM, EMD Millipore Corp) (HTF-BSA) covered with 4 ml of mineral oil (Cat# M8410-500ML, Sigma). Three incisions were made on the cauda to allow sperm to swim out and to get capacitated for at least 30 min. 25 µl of mTmG and 25 µl quinKO sperm suspensions were mixed, and 40 µl (if using 500 µl HTF-BSA) or 25 µl (if using 150 µl HTF-BSA) of the mixed sperm were delivered to superovulated females using C&I Device for Mice (Cat# 60020, Paratech) at 9:00 a.m. Recipients were immediately paired with vasectomized males overnight. The next day, the plug was checked and the female mice with plugs were used for collecting embryonic day 10 (E10) embryos, and the ones without plugs were used for collecting zygotes, two-cell embryos, morulae, or blastocysts embryos.

## Mouse genotyping

Mouse tail snips were lysed in a lysis buffer (40 mM NaOH, 0.2 mM EDTA, pH = 12) for 1 hr at 95°C, followed by neutralization with the same volume of neutralizing buffer (40 mM Tris–HCl, pH 5.0). PCR reactions were conducted using the 2×GoTaq Green master mix (Promega, Cat# M7123). The primers used for genotyping are the same as previously described (*Wang et al., 2020b*). For single embryo genotyping (e.g., zygotes, two-cell embryos, four-cell embryos, morulae, and blastocysts), each embryo was picked up by mouse pipetting and transferred into a 200 µl tube, and lysed in 10 µl of lysis buffer (100 mM Tris–HCl [pH 8.0], 100 mM KCl, 0.02% gelatin, 0.45% Tween 20, 60 µg/ml yeast tRNA, and 125 µg/ml proteinase K) at 55°C for 30 min followed by inactivation at 95°C for 10 min. 2 µl of the lysis was used as the template for the first round of PCR (30 cycles) in a 10 µl reaction using the PrimeSTAR HS DNA Polymerase (Cat# R010B, Takara) or 2×GoTaq Green Master Mix (Cat# M7123, Promega). Then, 2 µl of the first PCR was used for the second round of PCR in a 10 µl reaction using 2×GoTaq Green Master Mix (Cat# M7123, Promega) for 35 cycles. Primers used for embryo geno-typing are included in *Supplementary file 8*.

## openCASA

Sperm parameters were assessed using openCASA (*Alquézar-Baeta et al., 2019*). After sperm capacitation in HTF containing 4% BSA at 37°C for 30 min, the video was recorded as an AVI format at 60 frames per second (FPS) for 2 s with a resolution of 768 * 576 pixels using UPlan FL N 4×/0.13 PhP Objective Lens (Olympus) and DMK 33UP1300 camera (The Imaging Source). Motility module in openCASA was set with the following parameters: 1.21 microns per pixel, the cell size of 10–200 µm$^2$, progressive motility (STR > 50%, VAP > 50), minimum VCL of 10 µm/s, VCL threshold of 30–200 µm/s, 60 frame rate (frames/s), 10 minimum track length (frames), 20 µm maximum displacement between frames, and window size (frames) of 4.

## Analysis of conservation of *MIR-506* family in modern humans using the 1000 Genomes Project (1kGP)

The vcf files from the 1000 Genomes Project covering 3202 samples were downloaded from here. The miRNA annotations were obtained from UCSC genome browser, and pachytene piRNA hg19 genome coordinance was obtained from *Özata et al., 2020* and converted to GRCh38 genome coordinance using LiftOver. The DAF was retrieved from the vcf file, and mean nucleotide diversity (MND) was calculated as 2 * DAF * (1-DAF) as previously described (*Özata et al., 2020*). Kruskal–Wallis test was used for statistical analysis, and adjusted p<0.05 was identified as statistically significant.

## Overexpression of MER91C

RNA structures for MER91C were predicted using RNAfold (*Lorenz et al., 2011*). MER91C DNA transposons from humans (*MIR513A1*), dogs (*MIR507B*), and horses (*MIR514A*) were synthesized by IDT and inserted into pCI-Neo plasmid (Cat# E1841, Promega) using NEBuilder HiFi DNA Assembly Master Mix (Cat# E2621L, NEB). 150 ng of pCI-Neo (negative control) or pCI-Neo-MER91C were transfected with or without 150 ng of pcDNA3.1+_FH-AGO2-WT (Plasmid # 92006, Addgene) into HEK293T cells (Cat# CRL-3216, ATCC, RRID:CVCL_0063) at ~60% confluency in 24-well plates. 24 hr later, cells were harvested followed by RNA extraction using mirVana miRNA Isolation Kit (Cat# AM1561, Thermo Fisher Scientific), polyadenylation by *E. coli* Poly(A) Polymerase (Cat# M0276L, NEB), reverse transcription using SuperScript IV Reverse Transcriptase (Cat# 18090010, Thermo Fisher Scientific), and PCR using 2×GoTaq Green Master Mix (Cat# M7123, Promega) or qPCR by PowerUp SYBR Green Master Mix (Cat# A25742, Thermo Fisher Scientific). Primers used for RT-PCR, PCR, and qPCR are included in *Supplementary file 8*.

## Bioinformatic analyses of transposable element (TE)

Genomic regions and GFF3 files for transcript 3'UTR annotations were downloaded from the UCSC genome browser, and GTF files for transposon annotations were downloaded from here. For the TE containing transcripts, bedtools closest was used to extract the closest TEs to transcripts. The genomes used were GRCh38 (humans) and GRCm39 (mice).

## Phylogenetic tree analysis of the MER91C DNA transposon and the *MIR-506* family miRNAs

The MIR-506 family miRNA sequences were retrieved from the MirGeneDB (https://mirgenedb.org/; humans, rhesus monkeys, rats, guinea pigs, rabbits, tenrecs, and cows), miRBase (horses), or UCSC genome browser (marmoset monkeys and green sea turtles). The transposon fasta sequences from humans, dogs, horses, and guinea pigs were downloaded from the UCSC genome browser and aligned to the *MIR-506* family miRNAs in their corresponding species using BLAST (*Altschul et al., 1990*). After retrieving the transposons that aligned to the *MIR-506* family miRNAs, the *MIR-506* family miRNAs and the transposons were aligned using ClustalW2 followed by phylogenetic tree building using IQ-TREE2 with default parameters (*Minh et al., 2020*). The final figure was generated using Geneious software.

## Purification of germ cells

Pachytene spermatocytes and round spermatids were purified from adult C57BL/6J mice using the STA-PUT method. BSA gradients (2–4%) were prepared in EKRB buffer with a pH of 7.2 containing 1×

Krebs-Ringer Bicarbonate Buffer (Cat# K4002, Sigma), 1.26 g/l sodium bicarbonate (Cat# S6761, Sigma), 1× GlutaMAX (Cat# 35050061, Thermo Fisher Scientific), 1× Antibiotic-Antimycotic (Cat# 15240062, Thermo Fisher Scientific), 1× MEM Non-Essential Amino Acids (Cat# 11140050, Thermo Fisher Scientific), 1× MEM Amino Acids (Cat# 11130051, Thermo Fisher Scientific), and 100 ng/ml cycloheximide (Cat# 01810, Sigma). After being removed and decapsulated, testes were placed into 10 ml of EKRB buffer containing 0.5 mg/ml type IV collagenase (Cat# C5138, Sigma) and digested at 33°C for ~12 min to dissociate the seminiferous tubules. Once dissociated, the seminiferous tubules were washed three times using EKRB buffer to remove the interstitial cells and red blood cells followed by trypsin digestion by incubation at 33°C for ~12 min with occasional pipetting in 10 ml EKRB buffer containing 0.25 mg/ml trypsin (Cat# T9935, Sigma) and 20 µg/ml DNase I (Cat# DN25, Sigma). 1 ml of 4% BSA-EKRB was added to the 10 ml fully dispersed testicular cells to neutralize the trypsin digestion followed by centrifuge at 800 × $g$ for 5 min at 4°C. Testicular cells were washed two times with EKRB buffer and resuspended in 10 ml 0.5% BSA-EKRB. The cell suspension was passed through 70 µm cell strainer (Cat# 431751, Corning) and loaded onto the STA-PUT apparatus containing 2–4% BSA-EKRB gradients for sedimentation. After 2–3 hr sedimentation at 4°C, cell fractions were collected from the bottom of the sedimentation chamber. Fractions containing the same cell types were pooled and saved for RNA sequencing.

## Library construction and RNA-seq

RNA was extracted using the mirVana miRNA Isolation Kit (Cat# AM1561, Thermo Fisher Scientific) following the manufacturer's instructions. Large RNA (>200 nt) and small RNA (<200 nt) were isolated separately for library construction. Small RNA libraries were constructed using NEBNext Small RNA Library Prep Set for Illumina (Multiplex Compatible) (Cat# E7330L, NEB) following the manufacturer's instructions, and sequenced using HiSeq 2500 system for single-end 50 bp sequencing. Large RNA libraries were constructed using the KAPA Stranded RNA-Seq Kit with RiboErase (Cat# KK8483, Roche) and the adaptor from NEBNext Multiplex Oligos for Illumina (Index Primers Set 1, Cat# E7335L, NEB). The indexed large RNA libraries were sequenced using Nextseq 500 with paired-end 75 bp sequencing.

## Large and small RNA-seq data analysis

For the large RNA-seq data, raw sequences were trimmed by Trimmomatic (*Bolger et al., 2014*), followed by alignment using Hisat2 (*Pertea et al., 2016*), and assembly using StringTie (*Pertea et al., 2016*). Reads were summarized using featureCounts (*Liao et al., 2014*) and the differential gene expression was compared using DESeq2 (*Love et al., 2014*). For each KO mouse sample, the genes with a fold change ≥2 and FDR < 0.05 were considered DEGs. The DEGs in each KO mouse were then intersected with the corresponding miRNA targets predicted by four different algorithms, including TargetScan (*Agarwal et al., 2015*), microrna.org (*Betel et al., 2010*), miRWalk (*Dweep and Gretz, 2015*), and mirDB (*Chen and Wang, 2020*). The gene is considered a putative target (DETs) as long as it intersects with the targets predicted by any method mentioned above. The DETs identified in each KO mouse sample were intersected with other KO mouse samples, and the DETs intersected at least two different KO mouse samples were selected as the 'pool' of DETs in mice. Then the DETs 'pool' in mice was intersected with *MIR-506* family predicted targets in humans and rats to determine the shared targets among humans, rats, and mice. A mRNA that any *MIR-506* family member is targeting is deemed as the shared target.

For the small RNA-seq data, we applied the AASRA (*Tang et al., 2021*) pipeline (for mice) or SPORTS1.0 (*Shi et al., 2018*) (for humans, monkeys, rats, and horses) to parse the raw sequencing data. The clean reads were mapped against miRbase (*Kozomara et al., 2019*) or MirGeneDB (*Fromm et al., 2020*). Due to the ill-annotated marmoset miRNA reference, we used the rhesus monkey miRNA reference for the marmoset miRNA alignment and assigned the marmoset miRNA names based on the rhesus monkey miRNA reference. The DESeq2 (*Love et al., 2014*) (for mice) or edgeR (*Robinson et al., 2010*) (for humans, monkeys, rats, and horses) algorithm was used to compare the groupwise miRNA expression levels. The RNAs with an FDR < 5% were deemed differentially expressed. Cohen's $d$ was computed by the 'cohensD' function within the 'lsr' package.

## Cell line

HEK293T cells were ordered from ATCC (Cat# CRL-3216, RRID:CVCL_0063) with no mycoplasma contaminations detected.

## Luciferase assay

For luciferase reporter assays, the 3' UTR of *Crisp1*, *Egr1*, *hCRISP1*, and *hFMR1* were amplified using C57BL/6J tail snips or HEK293 cells genomic DNA template with Q5 Hot Start High-Fidelity 2X Master Mix (Cat# M0494L, NEB). The PCR products were inserted into psiCHECK-2 vector (Cat# C8021, Promega) via Xho I (Cat# R0146S, NEB) and Not I (Cat# R3189S, NEB) restriction enzymes cutting sites downstream of the Renilla luciferase-coding sequence using either NEBuilder HiFi DNA Assembly Master Mix (Cat# E2621L, NEB) or T4 DNA ligase (Cat# M0202L, NEB). For the miRNA over-expression plasmids, ~300 bp upstream and downstream of the precursor miRNA genomic region were amplified using C57BL/6J tail snips or HEK293T cells (Cat# CRL-3216, ATCC, RRID:CVCL_0063) genomic DNA with Q5 Hot Start High-Fidelity 2X Master Mix (Cat# M0494L, NEB) or PrimeSTAR HS DNA Polymerase (Cat# R010B, Takara), and inserted into pcDNA3.1 plasmids using NEBuilder HiFi DNA Assembly Master Mix (Cat# E2621L, NEB). The cloned products were introduced into Mix & Go competent cells (DH5 alpha strain, Cat# T3007, Zymo Research) for transformation, followed by positive colonies picking, sequencing, and plasmids extraction. HEK293T (Cat# CRL-3216, ATCC, RRID:CVCL_0063) cells were co-transfected with 150 ng pcDNA3.1-miRNA and 150 ng psiCHECK-2 containing the 3'UTR of the target gene using Lipofectamine 3000 (Cat# L3000015, Thermo Fisher Scientific) in a 24-well cell culture plate (Cat# 3524, Corning) at ~60% confluency. After 24 hr of culture, cells were lysed and assayed with Dual Luciferase Assays (Cat# E1910, Promega) according to the manufacturer's instructions. Renilla luciferase signals were normalized to Firefly luciferase signals to adjust the transfection efficiency. pcDNA3.1-cel-mir-67, which has a minimal sequence identity to the miRNAs in humans, mice, and rats, was used as a negative control miRNA. Primers used for generating plasmids containing miRNAs or 3'UTR of the target genes are included in *Supplementary file 8*.

## Immunofluorescence

Cauda sperm were capacitated in HTF at 37°C for half an hour followed by spreading onto Superfrost Plus slides (Cat# 22-037-246, Thermo Fisher Scientific). The slides were air-dried, fixed in 4% paraformaldehyde for 15 min (Cat# J19943-K2, Thermo Fisher Scientific), then washed twice in 0.4% Photo-Flo 200 (Cat# 1464510, Kodak) /1×PBS (5 min/wash), followed by a 5 min wash in 0.4% Photo-Flo 200/ ddH$_2$O, and stored in –80°C after air-dried. The slides were equilibrated to room temperature before immunofluorescence, followed by incubation in acetone for 20 min at 4°C and rehydration in 95% ethanol twice (5 min/wash), 70% ethanol twice (5 min/wash), and 1× PBS for three times (5 min/ wash) sequentially. Heat-induced antigen retrieval was performed in citrate buffer (pH 6.0) with high power for 4 min once, and three times with low power for 4 min in microwave. Slides were cooled down to room temperature and washed in 1× PBS twice (5 min/wash). Following permeabilization with 0.25% Triton X-100 (Sigma-Aldrich, Cat# T8787) in 1× PBS for 20 min at room temperature, the slides were washed with 1× PBS three times (5 min/wash) and incubated at 3% H$_2$O$_2$ solution to block endogenous peroxidase activity. After washing in 1× PBS twice (5 min/wash), the slides were blocked with 1× blocking solution (5% normal donkey serum, 5% fetal bovine serum, and 1% bovine serum albumin in 1× PBS) at room temperature for 1 hr, then incubated with the anti-CRISP-1 antibody (Cat# AF4675-SP, R&D Systems, RRID:AB_2687670, 1:100 in 1× blocking solution) at 4°C overnight. After primary antibody incubation, the slides were washed in 1× PBS three times (10 min/wash), followed by incubation in the donkey anti-goat IgG H&L (HRP) (Cat# ab97110, Abcam, 1:250 in 1× blocking solution) at room temperature for 1 hr, and three times washes in 1× PBS (10 min/wash). Tyramide signal amplification was performed and stopped using reagents from Invitrogen Alexa Fluor 488 Tyramide SuperBoost Kit (Cat# B40941, Thermo Fisher Scientific), followed by mounting and counterstaining in Antifade Mounting Medium with DAPI (Cat# H-1800, Vector Lab). Nail polish was applied on the edge of the coverslips after 2 hr of mounting to prevent further evaporation and stored at 4°C before taking images. Images were taken using the Nikon ECLIPSE Ti2 Confocal microscope with the NIS-Elements Software.

## Western blot

Testes from adult WT and KO mice were collected and sonicated in 2× Laemmli buffer (Cat# 1610737, Bio-Rad) supplemented with 2-mercaptoethanol (Cat# M6250, Sigma-Aldrich) and cOmplete, Mini, EDTA-free Protease Inhibitor Cocktail (Cat# 11836170001, Sigma-Aldrich) followed by incubating at 100°C for 10 min. The proteins were separated on 4–20% Mini-PROTEAN TGX Precast Protein Gels

(Cat# 4561094, Bio-Rad) and then transferred onto Amersham Protran Premium Western blotting membranes, nitrocellulose (Cat# GE10600003, Sigma-Aldrich). The membranes were then stained with Ponceau S solution (Cat# P7170, Sigma-Aldrich) to check the samples' loading. After taking pictures, the membrane was destained with 0.1 M NaOH, and washed with water and TBS. Then the membrane was blocked with 5% skim milk in TBST (TBS containing 0.1% [v/v] Tween-20) for 1 hr at room temperature and incubated with the anti-CRISP-1 antibody (Cat# AF4675-SP, R&D Systems, RRID:AB_2687670, 1:2000 in TBST containing 5% skim milk) and anti-GAPDH antibody (Cat# G9545, Sigma, RRID:AB_796208, 1:6000 in TBST) overnight at 4°C. After washing with TBST three times, the membrane was incubated with the donkey anti-goat IgG H&L (HRP) (Cat# ab97110, Abcam) or goat anti-rabbit IgG H&L (HRP) (Cat# ab6721, Abcam) at room temperature for 1 hr. Followed by three washes with TBST, the bands were detected using the WesternBright ECL kit (Cat# K-12045-D20, Advansta).

## T7 endonuclease I (T7EI) assay

The potential off-target sites that may be induced by CRISPR-Cas9 were predicted using Alt-R Custom Cas9 crRNA Design Tool (IDT) and assessed by T7 endonuclease I (Cat# M0302L, NEB) assay. The sequences were retrieved from the UCSC genome browser, and the primers flanking the off-target sites were designed to cover ~600 bp. Genomic DNA from WT C57BL/6J or quinKO was amplified using Q5 Hot Start High-Fidelity 2X Master Mix (Cat# M0494L, NEB) or PrimeSTAR HS DNA Polymerase (Cat# R010B, Takara) with the designed off-target primers. 2 µl of the unpurified PCR product was diluted in 1× NEBuffer 2 in a 9.5 µl volume and denatured at 95°C for 5 min, followed by annealing at 95–85°C at a –2°C/s rate, and 85–25°C at a –0.1 °C/s rate. Then 0.5 µl of T7EI was added to the 9.5 µl denatured PCR products and incubated at 37°C for 30 min. The T7EI-treated PCR products were run on 1× TBE gel and stained with SYBR Gold Nucleic Acid Gel Stain (Cat# S11494, Thermo Fisher Scientific) to detect the off-target effects. Primers used for T7EI are included in **Supplementary file 8**.

## miRNA and 3′UTR conservation analysis

The Multiz Alignment and Conservation method was used to measure miRNA sequence conservation with the human genome as the reference (*Casper et al., 2018*; *Blanchette et al., 2004*). One hundred species were analyzed. PhastCons considers the flanking sequences and does not rely on fixed sliding windows; consequently, both highly conserved short sequences and moderately conserved long sequences can yield higher scores (*Siepel et al., 2005*). PhastCons gives a value between 0–1, the higher the value is, the more conserved the region is. By contrast, PhyloP compares the conservation of individual nucleotides among all phylogeny clades, giving positive scores once the region is conserved and vice versa. PhyloP and PhastCons scores of all miRNAs, *MIR-506* family, *FMR1* CDS, *SLITRK2* CDS, and the intergenic region (IGR) were retrieved from the UCSC genome browser and quantified. Kruskal–Wallis test was used for statistical analysis, and adjusted p<0.05 was identified as statistically significant. PhyloP scheme was used to measure the evolutionary conservation level at individual nucleotide sites in the 3′UTRs of the target genes. Positive PhyloP scores suggest higher conservation and stronger purifying selection, whereas negative PhyloP scores indicate accelerated evolution and potential adaptive selection. The genomic annotations and mRNA sequences were based on the hg38 (*Casper et al., 2018*) ('TxDb.Hsapiens.UCSC.hg38.knownGene' and 'BSgenome. Hsapiens.UCSC.hg38') and mm10 ('TxDb.Mmusculus.UCSC.mm10.knownGene' and 'BSgenome. Mmusculus.UCSC.mm10') assemblies for human and mouse, respectively. The PhyloP scores were mapped to individual nucleotides in 3′UTRs based on the transcript coordinates (*Lawrence et al., 2013*).

## Materials availability statement

Unique materials generated in this study are available from the corresponding author upon reasonable request.

## Statistical analyses

Data are presented as mean ± SEM, and statistical differences between datasets were assessed by two-samples *t*-test, *F*-test, Dunnett's multiple comparisons test as the post hoc test following

one-way ANOVA, chi-squared test, or Kruskal–Wallis test as described in the text or figure legends. Normal distribution was assessed by quantile-quantile (QQ) plot or density plot. $p < 0.05$, 0.01, 0.001, and 0.0001 are considered as statistically significant and indicated with *, **, ***, and **** respectively.

## Acknowledgements

We would like to thank Dr. Kevin J Peterson, Dartmouth College, Hanover, NH, for his help with phylogenetic analyses of the *MIR-506* family miRNA genes. This work was supported by grants from the NIH (HD071736, HD085506, HD098593, HD099924, and P30GM110767 to WY), NIH/ National Center for Advancing Translational Science (NCATS) UCLA CTSI (UL1TR001881-01 to ZW and WY), and the Templeton Foundation (PID: 61174 to WY). Work in the Lai lab was supported by the NIH (R01-GM083300 and R01-HD108914) and Memorial Sloan-Kettering Institute Grant P30-CA008748.

## Additional information

### Competing interests

Wei Yan: Senior editor, *eLife*. The other authors declare that no competing interests exist.

### Funding

| Funder | Grant reference number | Author |
| --- | --- | --- |
| Eunice Kennedy Shriver National Institute of Child Health and Human Development | HD071736 | Wei Yan |
| National Institute of General Medical Sciences | GM110767 | Wei Yan |
| National Center for Advancing Translational Sciences | UL1TR001881 | Zhuqing Wang Wei Yan |
| John Templeton Foundation | 61174 | Wei Yan |
| National Institute of General Medical Sciences | GM083300 | Eric C Lai |
| Eunice Kennedy Shriver National Institute of Child Health and Human Development | HD108914 | Eric C Lai |
| Memorial Sloan-Kettering Institute | CA008748 | Eric C Lai |
| Eunice Kennedy Shriver National Institute of Child Health and Human Development | HD085506 | Wei Yan |
| Eunice Kennedy Shriver National Institute of Child Health and Human Development | HD098593 | Wei Yan |
| Eunice Kennedy Shriver National Institute of Child Health and Human Development | HD099924 | Wei Yan |

The funders had no role in study design, data collection and interpretation, or the decision to submit the work for publication.

## Author contributions
Zhuqing Wang, Conceptualization, Resources, Data curation, Software, Formal analysis, Funding acquisition, Validation, Investigation, Visualization, Methodology, Writing – original draft, Project administration, Writing - review and editing; Yue Wang, Sheng Chen, Resources, Data curation, Validation, Methodology; Tong Zhou, Musheng Li, Data curation, Software, Methodology; Dayton Morris, Resources, Data curation, Validation, Investigation, Methodology; Rubens Daniel Miserani Magalhães, Data curation, Software, Formal analysis, Visualization, Methodology; Shawn Wang, Data curation, Validation, Investigation; Hetan Wang, Hayden McSwiggin, Data curation, Validation, Methodology; Yeming Xie, Resources, Data curation, Software, Methodology; Daniel Oliver, Shuiqiao Yuan, Huili Zheng, Resources, Methodology; Jaaved Mohammed, Data curation, Software, Visualization, Methodology; Eric C Lai, Data curation, Software, Investigation, Visualization, Methodology; John R McCarrey, Resources, Methodology, Writing – original draft; Wei Yan, Conceptualization, Resources, Data curation, Supervision, Funding acquisition, Validation, Investigation, Writing – original draft, Project administration, Writing - review and editing

## Author ORCIDs
Zhuqing Wang (iD) http://orcid.org/0000-0002-3988-0733
Shuiqiao Yuan (iD) http://orcid.org/0000-0003-1460-7682
Jaaved Mohammed (iD) http://orcid.org/0000-0002-7053-5575
Eric C Lai (iD) https://orcid.org/0000-0002-8432-5851
John R McCarrey (iD) https://orcid.org/0000-0002-5784-9318
Wei Yan (iD) http://orcid.org/0000-0001-9569-9026

## Ethics
All mice used in this study were on 2~3-month-old adult C57BL/6J background (Strain #:000664, The Jackson Laboratory, RRID: IMSR_JAX:000664) or CD1 background (Strain #:022, Charles River Laboratories, RRID:IMSR_CRL:022), and housed in a temperature- and humidity-controlled, specific pathogen-free facility under a light-dark cycle (12:12 light-dark) with food and water ad libitum. Animal use protocol was approved by the Institutional Animal Care and Use Committees (IACUC) of the University of Nevada, Reno (protocol: 00494) and The Lundquist Institute at Harbor-UCLA (protocol: 32132-03), and is following the "Guide for the Care and Use of Experimental Animals" established by the National Institutes of Health (1996, revised 2011).

Reviewer #1 (Public Review): https://doi.org/10.7554/eLife.90203.3.sa1
Reviewer #3 (Public Review): https://doi.org/10.7554/eLife.90203.3.sa2
Author response https://doi.org/10.7554/eLife.90203.3.sa3

# Additional files

## Supplementary files
• Supplementary file 1. The X-linked *MIR-506* family miRNAs in different species.
• Supplementary file 2. TEs analysis for the X-linked *MIR-506* family miRNAs.
• Supplementary file 3. sRNA-seq analysis among different species.
• Supplementary file 4. Male infertility associated with the X-linked *MIR-506* family miRNAs in humans.
• Supplementary file 5. Dysregulated large RNAs in the X-linked *MIR-506* family miRNAs KO mice.
• Supplementary file 6. Dysregulated small RNAs in the X-linked *MIR-506* family miRNAs KO mice.
• Supplementary file 7. Common targets of the X-linked *MIR-506* family miRNAs among humans, mice, and rats.
• Supplementary file 8. Primers used in this study.
• MDAR checklist

## Data availability
The sRNA-seq and RNA-seq datasets have been deposited into the SRA database with accession#: PRJNA558973 and PRJNA670945. The scripts for sRNA analysis can be found on GitHub (https://

github.com/biogramming/AASRA; copy archived at *Biogramming, 2022* and https://github.com/junchaoshi/sports1.1; copy archived at *Junchaoshi, 2023*).

The following datasets were generated:

| Author(s) | Year | Dataset title | Dataset URL | Database and Identifier |
|---|---|---|---|---|
| Wang Z, Xie Y, Wang Y, Morris D, Wang S, Oliver D, Yuan S, Zayac K, Bloomquist S, Zheng H, Yan W | 2020 | X-linked miR-506 family miRNAs stabilize FMRP in mouse spermatogonia | https://www.ncbi.nlm.nih.gov/bioproject/PRJNA558973/ | NCBI BioProject, PRJNA558973 |
| Wang Z, Wang Y, Zhou T, Chen S, Morris D, Magalhães RDM, Li M, Wang S, Wang H, Xie Y, McSwiggin H, Oliver D, Yuan S, Zheng H, Mohammed J, Lai EC, McCarrey JR, Yan W | 2020 | Roles of X linked miR-506 family during spermatogenesis | https://www.ncbi.nlm.nih.gov/bioproject/PRJNA670945/ | NCBI BioProject, PRJNA670945 |

The following previously published datasets were used:

| Author(s) | Year | Dataset title | Dataset URL | Database and Identifier |
|---|---|---|---|---|
| Keller et al. | 2020 | Small non-coding RNA organ expression atlas | https://www.ncbi.nlm.nih.gov/bioproject/PRJNA686442/ | NCBI BioProject, PRJNA686442 |
| Bushel et al. | 2016 | microRNA profiling of Sprague Dawley organs | https://www.ncbi.nlm.nih.gov/search/all/?term=PRJNA312384 | NCBI BioProject, PRJNA312384 |
| Koenig et al. | 2016 | The Beagle Dog MicroRNA Tissue Atlas: Identifying Translatable Biomarkers of Organ Toxicity | https://www.ncbi.nlm.nih.gov/bioproject/PRJNA325490/ | NCBI BioProject, PRJNA325490 |
| Gainetdinov et al. | 2016 | Human small RNAs (miRNAs, piRNAs, 5'-tiRs) in postnatal testis, carcinoma in situ (CIS), and testicular germ cell tumors | https://www.ncbi.nlm.nih.gov/bioproject/PRJNA352412/ | NCBI BioProject, PRJNA352412 |
| Li B, He X | 2017 | Transcriptome Analysis of piRNAs during testicular development and spermatogenesis of Mongolian horse | https://www.ncbi.nlm.nih.gov/geo/query/acc.cgi?acc=GSE100852 | NCBI Gene Expression Omnibus, GSE100852 |
| Hirano T, Iwasaki YW, Siomi H | 2014 | Small RNA and gene expression profile in the adult testes of the common marmoset | https://www.ncbi.nlm.nih.gov/geo/query/acc.cgi?acc=GSE52927 | NCBI Gene Expression Omnibus, GSE52927 |

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
