## [Editor Report · eLife assessment]

This study provides **important** findings on the evolution and function of the X-linked *MIR-506* family. The evidence supporting the conclusions is **convincing**, including the generation and characterization of an impressive number of the miRNA deletion mutants. This work will be of interest to RNA biologists, evolution biologists, and reproductive biologists.

---

## [Referee Report · Reviewer #1 (Public Review)]

Wang et al investigated the evolution, expression, and function of the X-linked miR-506 miRNA family. They showed that the miR-506 family underwent rapid evolution. They provided evidence that miR-506 appeared to have originated from the MER91C DNA transposons. Human MER91C transposon produced mature miRNAs when expressed in cultured cells. A series of mouse mutants lacking individual clusters, a combination of clusters, and the entire X-linked cluster (all 22 miRNAs) were generated and characterized. The mutant mice lacking four or more miRNA clusters showed reduced reproductive fitness (litter size reduction). They further showed that the sperm from these mutants were less competitive in polyandrous mating tests. RNA-seq revealed the impact of deletion of miR-506 on the testicular transcriptome. Bioinformatic analysis analyzed the relationship among miR-506 binding, transcriptomic changes, and target sequence conservation. The miR-506-deficient mice did not have apparent effect on sperm production, motility, and morphology. Lack of severe phenotypes is typical for miRNA mutants in other species as well. However, the miR-506-deficient males did exhibit reduced litter size, such an effect would have been quite significant in an evolutionary time scale. The number of mouse mutants and sequencing analysis represent a tour de force.

Strengths:

This study is a comprehensive investigation of the X-linked miR-506 miRNA family. It provides important insights into the evolution and function of the miR-506 family.

---

## [Referee Report · Reviewer #3 (Public Review)]

Summary:

In this manuscript, the authors conducted a comprehensive study of the X-linked miR-506 family miRNAs in mice on its origin, evolution, expression, and function. They demonstrate that the X-linked miR-506 family, predominantly expressed in the testis, may be derived from MER91C DNA transposons and further expanded by retrotransposition. By genetic deletion of different combinations of 5 major clusters of this miRNA family in mice, they found these miRNAs are not required for spermatogenesis. However, by further examination, the mutant mice show mild fertility problem and inferior sperm competitiveness. The authors conclude that the X-linked miR-506 miRNAs finetune spermatogenesis to enhance sperm competition.

Strengths:

This is a comprehensive study with extensive computational and genetic dissection of the X-linked miR-506 family providing a holistic view of its evolution and function in mice. The finding that this family miRNAs could enhance sperm competition is interesting and could explain their roles in finetuning germ cell gene expression to regulate reproductive fitness.

Comments on revised version:

The authors have addressed the concerns raised.

---

## [Author Response]

The following is the authors’ response to the original reviews.

**Reviewer #1 (Public Review):**
Wang et al investigated the evolution, expression, and function of the X-linked miR-506 miRNA family. They showed that the miR-506 family underwent rapid evolution. They provided evidence that miR-506 appeared to have originated from the MER91C DNA transposons. Human MER91C transposon produced mature miRNAs when expressed in cultured cells. A series of mouse mutants lacking individual clusters, a combination of clusters, and the entire X-linked cluster (all 22 miRNAs) were generated and characterized. The mutant mice lacking four or more miRNA clusters showed reduced reproductive fitness (litter size reduction). They further showed that the sperm from these mutants were less competitive in polyandrous mating tests. RNA-seq revealed the impact of deletion of miR-506 on the testicular transcriptome. Bioinformatic analysis analyzed the relationship among miR-506 binding, transcriptomic changes, and target sequence conservation. The miR-506-deficient mice did not have apparent effect on sperm production, motility, and morphology. Lack of severe phenotypes is typical for miRNA mutants in other species as well. However, the miR-506-deficient males did exhibit reduced litter size, such an effect would have been quite significant in an evolutionary time scale. The number of mouse mutants and sequencing analysis represent a tour de force. This study is a comprehensive investigation of the X-linked miR-506 miRNA family. It provides important insights into the evolution and function of the miR-506 family.The conclusions of this preprint are mostly supported by the data except being noted below. Some descriptions need to be revised for accuracy.L219-L285: The conclusion that X-linked miR-506 family miRNAs are expanded via LINE1 retrotransposition is not supported by the data. LINE1s and SINEs are very abundant, accounting for nearly 30% of the genome. In addition, the LINE1 content of the mammalian X chromosome is twice that of the autosomes. One can easily find flanking LINE1/SINE repeat. Therefore, the analyses in Fig. 2G, Fig. 2H and Fig. S3 are not informative. In order to claim LINE1-mediated retrotransposition, it is necessary to show the hallmarks of LINE1 retrotransposition, which are only possible for new insertions. The X chromosome is known to be enriched for testis-specific multi-copy genes that are expressed in round spermatids (PMID: 18454149). The conclusion on the LINE1-mediated expansion of miR-506 family on the X chromosome is not supported by the data and does not add additional insights. I think that the LINE1 related figure panels and description (L219-L285) need to be deleted. In discussion (L557558), "...and subsequently underwent sequence divergence via LINE1-mediated retrotransposition during evolution" should also be deleted. This section (L219-L285) needs to deal only with the origin of miR506 from MER91C DNA transposons, which is both convincing and informative.

Reply: Agreed, the corresponding sentences were deleted.

Fig. 3A: can you speculate/discuss why the miR-506 expression in sperm is higher than in round spermatids?

Reply: RNAs are much less abundant in sperm than in somatic or spermatogenic cells (~1/100). Spermborne small RNAs represent a small fraction of total small RNAs expressed in their precursor spermatogenic cells, including spermatocytes and spermatids. Therefore, when the same amount of total/small RNAs are used for quantitative analyses, sperm-borne small RNAs (e.g., *MIR-506* family miRNAs) would be proportionally enriched in sperm compared to other spermatogenic cells. We discussed this point in the text (Lines 550-556).

**Reviewer #2 (Public Review):In this paper, Wang and collaborators characterize the rapid evolution of the X-linked miR-506 cluster in mammals and characterize the functional reference of depleting a few or most of the miRNAs in the cluster. The authors show that the cluster originated from the MER91C DNA transposon and provide some evidence that it might have expanded through the retrotransposition of adjacent LINE1s. Although the animals depleted of most miRNAs in the cluster show normal sperm parameters, the authors observed a small but significant reduction in litter size. The authors then speculate that the depletion of most miRNAs in the cluster could impair sperm competitiveness in polyandrous mating. Using a successive mating protocol, they show that, indeed, sperm lacking most X-linked miR-506 family members is outcompeted by wild-type sperm. The authors then analyze the evolution of the miR-506 cluster and its predicted targets. They conclude that the main difference between mice and humans is the expansion of the number of target sites per transcript in humans.The conclusions of the paper are, in most cases, supported by the data; however, a more precise and indepth analysis would have helped build a more convincing argument in most cases.(1) In the abstracts and throughout the manuscript, the authors claim that "... these X-linked miRNA-506 family miRNA [...] have gained more targets [...] " while comparing the human miRNA-506 family to the mouse. An alternative possibility is that the mouse has lost some targets. A proper analysis would entail determining the number of targets in the mouse and human common ancestor.

Reply: This question alerted us that we did not describe our conclusion accurately, causing confusion for this reviewer. Our data suggest that although the sheer number of target genes remains the same between humans and mice, the human X-linked *MIR-506* family targets a greater number of genes than the murine counterpart on a per miRNA basis. In other words, mice never lost any targets compared to humans, but per the *MIR-506* family miRNA tends to target more genes in humans than in mice.

We revised the text to more accurately report our data. The pertaining text (lines 490-508) now reads: “Furthermore, we analyzed the number of all potential targets of the *MIR-506* family miRNAs predicted by the aforementioned four algorithms among humans, mice, and rats. The total number of targets for all the X-linked *MIR-506* family miRNAs among different species did not show significant enrichment in humans (Figure 6-figure supplement 1 C), suggesting the sheer number of target genes does not increase in humans. We then compared the number of target genes per miRNA. When comparing the number of target genes per miRNA for all the miRNAs (baseline) between humans and mice, we found that on a per miRNA basis, human miRNAs have more targets than murine miRNAs (p<0.05, t-test) (Fig. S9D), consistent with higher biological complexity in humans. This became even more obvious for the X-linked miR-506 family (p<0.05, t-test) (Figure 6-figure supplement 1 D). In humans, the X-linked *MIR-506* family, on a per miRNA basis, targets a significantly greater number of genes than the average of all miRNAs combined (p<0.05, t-test) (Figure 6-figure supplement 1 D). In contrast, in mice, we observed no significant difference in the number of targets per miRNA between X-linked miRNAs and all of the mouse miRNAs combined (mouse baseline) (Figure 6-figure supplement 1 D). These results suggest that although the sheer number of target genes remains the same between humans and mice, the human X-linked *MIR-506* family targets a greater number of genes than the murine counterpart on a per miRNA basis.”

We also changed “have gained” to “have” throughout the text to avoid confusion.

(2) The authors claim that the miRNA cluster expanded through L1 retrotransposition. However, the possibility of an early expansion of the cluster before the divergence of the species while the MER91C DNA transposon was active was not evaluated. Although L1 likely contributed to the diversity within mammals, the generalization may not apply to all species. For example, SINEs are closer on average than L1s to the miRNAs in the SmiR subcluster in humans and dogs, and the horse SmiR subcluster seems to have expanded by a TE-independent mechanism.

Reply: Agreed. We deleted the data mentioned by this reviewer.

(3) Some results are difficult to reconcile and would have benefited from further discussion. The miR-465 sKO has over two thousand differentially expressed transcripts and no apparent phenotype. Also, the authors show a sharp downregulation of CRISP1 at the RNA and protein level in the mouse. However, most miRNAs of the cluster increase the expression of Crisp1 on a reporter assay. The only one with a negative impact has a very mild effect. miRNAs are typically associated with target repression; however, most of the miRNAs analyzed in this study activate transcript expression.

Reply: Both mRNA and protein levels of Crisp1 were downregulated in KO mice, and these results are consistent with the luciferase data showing overexpression of these miRNAs upregulated the Crisp1 3′UTR luciferase activity. We agree that miRNAs usually repress target gene expression. However, numerous studies have also shown that some miRNAs, such as human miR-369-3, Let-7, and miR-373, mouse miR-34/449 and the *MIR-506* family, and the synthetic miRNA miRcxcr4, activate gene expression both in vitro (1, 2) and in vivo (3-6). Earlier reports have shown that these miRNAs can upregulate their target gene expression, either by recruiting FXR1, targeting promoters, or sequestering RNA subcellular locations (1, 2, 6). We briefly discussed this in the text (Lines 605-611).

(4) More information is required to interpret the results of the differential RNA targeting by the murine and human miRNA-506 family. The materials and methods section needs to explain how the authors select their putative targets. In the text, they mention the use of four different prediction programs. Are they considering all sites predicted by any method, all sites predicted simultaneously by all methods, or something in between? Also, what are they considering as a "shared target" between mice and humans? Is it a mRNA that any miR-506 family member is targeting? Is it a mRNA targeted by the same miRNA in both species? Does the targeting need to occur in the same position determined by aligning the different 3'UTRs?

Reply: Since each prediction method has its merit, we included all putative targets predicted by any of the four methods. The "shared target" refers to a mRNA that any *MIR-506* family member targets because the *MIR-506* family is highly divergent among different species. We have added the information to the “Large and small RNA-seq data analysis” section in Materials and Methods (Lines 871-882).

(5) The authors highlight the particular evolution of the cluster derived from a transposable element. Given the tendency of transposable elements to be expressed in germ cells, the family might have originated to repress the expression of the elements while still active but then remained to control the expression of the genes where the element had been inserted. The authors did not evaluate the expression of transcripts containing the transposable element or discuss this possibility. The authors proposed an expansion of the target sites in humans. However, whether this expansion was associated with the expansion of the TE in humans was not discussed either. Clarifying whether the transposable element was still active after the divergence of the mouse and human lineages would have been informative to address this outstanding issue.

Reply: Agreed. The MER91C DNA transposon is denoted as nonautonomous (7); however, whether it was active during the divergence of mouse and human lineages is unknown. To determine whether the expansion of the target sites in humans was due to the expansion of the MER91C DNA transposon, we analyzed the MER91C DNA transposon-containing transcripts and associated them with our DETs. Of interest, 28 human and 3 mouse mRNAs possess 3′UTRs containing MER91C DNA sequences, and only3 and 0 out of those 28 and 3 genes belonged to DETs in humans and mice, respectively (Figure 6-figure supplement 1 E), suggesting a minimal effect of MER91C DNA transposon expansion on the number of target sites. We briefly discussed this in the text (Lines 511-518).

Post-transcriptional regulation is exceptionally complex in male haploid cells, and the functional relevance of many regulatory pathways remains unclear. This manuscript, together with recent findings on the role of piRNA clusters, starts to clarify the nature of the selective pressure that shapes the evolution of small RNA pathways in the male germ line.

Reply: Agreed. We appreciate your insightful comments.

**Reviewer #3 (Public Review):**
Summary:In this manuscript, the authors conducted a comprehensive study of the X-linked miR-506 family miRNAs in mice on its origin, evolution, expression, and function. They demonstrate that the X-linked miR-506 family, predominantly expressed in the testis, may be derived from MER91C DNA transposons and further expanded by retrotransposition. By genetic deletion of different combinations of 5 major clusters of this miRNA family in mice, they found these miRNAs are not required for spermatogenesis. However, by further examination, the mutant mice show mild fertility problem and inferior sperm competitiveness. The authors conclude that the X-linked miR-506 miRNAs finetune spermatogenesis to enhance sperm competition.Strengths:This is a comprehensive study with extensive computational and genetic dissection of the X-linked miR506 family providing a holistic view of its evolution and function in mice. The finding that this family miRNAs could enhance sperm competition is interesting and could explain their roles in finetuning germ cell gene expression to regulate reproductive fitness.Weaknesses:The authors specifically addressed the function of 5 clusters of X-link miR-506 family containing 19 miRNAs. There is another small cluster containing 3 miRNAs close to the Fmr1 locus. Would this small cluster act in concert with the 5 clusters to regulate spermatogenesis? In addition, any autosomal miR-506 like miRNAs may compensate for the loss of X-linked miR-506 family. These possibilities should be discussed.

Reply: The three FmiRs were not deleted in this study because the SmiRs are much more abundant than the FmiRs in WT mice (Author Response image 1, heatmap version of Figure 5C). Based on small RNA-seq, some FmiRs, e.g., miR-201 and miR-547, were upregulated in the SmiRs KO mice, suggesting that this small cluster may act in concert with the other 5 clusters and thus, worth further investigation. To our best knowledge, all the *MIR-506* family miRNAs are located on the X chromosome, although some other miRNAs were upregulated in the KO mice, they don’t belong to the *MIR-506* family. We briefly discussed this point in the text (Lines 635-638).

**Author response image 1. sa3fig1:** sRNA-seq of WT and *MIR-506* family KO testis samples.

Direct molecular link to sperm competitiveness defect remains unclear but is difficult to address.

Reply: In this study, we identified a target of the *MIR-506* family, i.e. Crisp1. KO of Crisp1 in mice, or inhibition of CRISP1 in human sperm (7, 8), appears to phenocopy the quinKO mice, displaying largely normal sperm motility but compromised ability to penetrate eggs. The detailed mechanism warrants further investigation in the future.

**Recommendations for the authors:**

**Reviewer #1 (Recommendations For The Authors):**
Lines 84-85: "Several cellular events are unique to the male germ cells, e.g., meiosis, genetic recombination, and haploid male germ cell differentiation (also called spermiogenesis)". This statement is not accurate. Please revise. Meiosis and genetic recombination are common to both male and female germ cells. They are highly conserved in both sexes in many species including mouse.

Reply: Agreed. We have revised the sentence and it now reads: “Several cellular events are unique to the male germ cells, e.g., postnatal formation of the adult male germline stem cells (i.e., spermatogonia stem cells), pubertal onset of meiosis, and haploid male germ cell differentiation (also called spermiogenesis) (9)” (Lines 83-86).

Lines 163-164: "we found that Slitrk2 and Fmr1 were syntenically linked to autosomes in zebrafish and birds (Fig. 1A), but had migrated onto the X chromosome in most mammals". This description is not accurate. Chr 4 in zebrafish and birds is syntenic to the X chromosome in mammals. The term "migrated" is not appropriate. Suggestion: Slitrk2 and Fmr1 mapped to Chr 4 (syntenic with mammalian X chromosome) in zebrafish and birds but to the X chromosome in most mammals.

Reply: Agreed. Revised as suggested.

**Reviewer #2 (Recommendations For The Authors):**
(1) In the significance statement, the authors mention that the mutants are "functionally infertile," although the decrease in competitiveness is partial. I suggest referring to them as "functionally sub-fertile."

Reply: Agreed. Revised as suggested.

(2) I will urge the authors to explain in more detail how some figures are generated and what they mean. Some critical information needs to be included in various panels.(2a) Figure S1. The phastCons track does not seem to align as expected with the rest of the figure. The highest conservation peak is only present in humans, and the sequence conserved in the sea turtle has the lowest phastCons score. I was expecting the opposite from the explanation.

Reply: The tracks for phyloP and phastCons are the scores for all 100 species, whereas the tracks with the species names on the left are the corresponding sequences aligned to the human genome. We have revised our figure to make it clearer.

(2b) Figure 2A and Figure S2C. Although all the functional analysis of the manuscript has been done in mice, the alignments showing sequence conservation do not include the murine miRNAs. Please include the mouse miRNAs in these panels.

Reply: The mouse has Mir-506-P7 with the conserved miRNA-3P seed region, which was included in the lower panel in Figure 1-figure supplement 2 C. However, mice do not have Mir-506-P6, which may have been lost or too divergent to be recognized during the evolution and thus, were not included in Figure 2A and the upper panel in Figure 1-figure supplement 2 C.

(2c) Figure S7H. The panel could be easier to read.

Reply: Agreed. We combined all the same groups and turned Figure S7H (now Figure 3-figure supplement 1 H) into a heatmap.

(2d) The legend of Figure 6G reads, "The number of target sites within individual target mRNAs in both humans and mice ." Can the author explain why the value 1 of the human "Number of target sites" is connected to virtually all the "Number of target sites" values in mice?

Reply: Sorry for the confusion. For example, for gene 1, we have 1 target site in the human and 1 target site in the mouse; but for gene 2, we have 1 target site in the human and multiple sites in the mouse; therefore, the value 1 is connected to more than one value in the mouse.

**Reviewer #3 (Recommendations For The Authors):**
CRISP1 and EGR1 protein localization in WT and mutant sperm by immunostaining would be helpful.

Reply: Agreed. We performed immunostaining for CRISP1 on WT sperm, and the new results are presented in Figure 5-figure supplement 1 D. CRISP1 seems mainly expressed in the principal piece and head of sperm.

The detailed description of the generation of various mutant lines should be included in the Methods.

Reply: We added more details on the generation of knockout lines in the Materials and Methods (686-701).

References:

(1) S. Vasudevan, Y. Tong, J. A. Steitz, Switching from repression to activation: microRNAs can upregulate translation. Science 318, 1931-1934 (2007).

(2) R. F. Place, L. C. Li, D. Pookot, E. J. Noonan, R. Dahiya, MicroRNA-373 induces expression of genes with complementary promoter sequences. Proc Natl Acad Sci U S A 105, 1608-1613 (2008).

(3) Z. Wang et al., X-linked miR-506 family miRNAs promote FMRP expression in mouse spermatogonia. EMBO Rep 21, e49024 (2020).

(4) S. Yuan et al., Motile cilia of the male reproductive system require miR-34/miR-449 for development and function to generate luminal turbulence. Proc Natl Acad Sci U S A 116, 35843593 (2019).

(5) S. Yuan et al., Oviductal motile cilia are essential for oocyte pickup but dispensable for sperm and embryo transport. Proc Natl Acad Sci U S A 118 (2021).

(6) M. Guo et al., Uncoupling transcription and translation through miRNA-dependent poly(A) length control in haploid male germ cells. Development 149 (2022).

(7) V. G. Da Ros et al., Impaired sperm fertilizing ability in mice lacking Cysteine-RIch Secretory Protein 1 (CRISP1). Dev Biol 320, 12-18 (2008).

(8) J. A. Maldera et al., Human fertilization: epididymal hCRISP1 mediates sperm-zona pellucida binding through its interaction with ZP3. Mol Hum Reprod 20, 341-349 (2014).

(9) L. Hermo, R. M. Pelletier, D. G. Cyr, C. E. Smith, Surfing the wave, cycle, life history, and genes/proteins expressed by testicular germ cells. Part 1: background to spermatogenesis, spermatogonia, and spermatocytes. Microsc Res Tech 73, 241-278 (2010).